# UniTST: Effectively Modeling Inter-Series and Intra-Series Dependencies for Multivariate Time Series Forecasting

**Juncheng Liu**[1]    **Chenghao Liu**[1]    **Gerald Woo**[1]    **Yiwei Wang**[2]
**Bryan Hooi**[3]    **Caiming Xiong**[1]    **Doyen Sahoo**[1]
*juncheng.liu@u.nus.edu*
*Salesforce*[1]    *University of California, Merced* [2]    *National University of Singapore*[3]

**Reviewed on OpenReview:** *https://openreview.net/forum?id=p3y5q4cvzV*

## Abstract

Transformer-based models have emerged as powerful tools for multivariate time series forecasting (MTSF). However, existing Transformer models often fall short of capturing both intricate dependencies across variate and temporal dimensions in MTS data. Some recent models are proposed to separately capture variate and temporal dependencies through either two sequential or parallel attention mechanisms. However, these methods cannot directly and explicitly learn the intricate inter-series and intra-series dependencies. In this work, we first demonstrate that these dependencies are very important as they usually exist in real-world data. To directly model these dependencies, we propose a transformer-based model UniTST containing a unified attention mechanism on the flattened patch tokens. Additionally, we add a dispatcher module which reduces the complexity and makes the model feasible for a potentially large number of variates. Although our proposed model employs a simple architecture, it offers compelling performance as shown in our extensive experiments on several datasets for time series forecasting.

## 1 Introduction

Inspired by the success of Transformer-based models in various fields such as natural language processing (Touvron et al., 2023a; Chiang et al., 2023; Almazrouei et al., 2023; MosaicML, 2023; Touvron et al., 2023b; OpenAI, 2022; Google, 2023; Touvron et al., 2023b) and computer vision (Wu et al., 2020; Liu et al., 2021b; Jamil et al., 2023), Transformers have also garnered much attention in the community of multivariate time series forecasting (MTSF) (Nie et al., 2023; Liu et al., 2024; Wu et al., 2021; Zhang & Yan, 2023; Zhou et al., 2022; Carlini et al., 2023; Han et al., 2024). Pioneering works (Li et al., 2021; Wu et al., 2021; Zhou et al., 2022) treat multiple variates (aka channels) at each time step as the input unit for transformers, similar to tokens in the language domain, but its performance was even inferior to linear models (Zeng et al., 2023; Han et al., 2023). Considering the noisy information from individual time points, *Variate-Independent* and *Patch-Based* (Nie et al., 2023) methods are subsequently proposed and achieve positive results by avoiding mixing noises from multiple variates and aggregating information from several adjacent time points as input. Nevertheless, these methods neglect the cross-variate relationships and interfere with the learning of temporal dynamics across variates.

To tackle this problem, iTransformer (Liu et al., 2024) embeds the entire time series of a variate into a token and employs "variate-wise attention" to model variate dependencies. However, it lacks the capability to model intra-variate temporal dependencies. Concurrently, several approaches (Zhang & Yan, 2023; Carlini et al., 2023; Yu et al., 2024) utilize both variate-wise attention and time(patch)-wise attention to capture inter-variate and intra-variate dependencies, either sequentially or parallelly. Yet, they may raise the difficulty of modeling the diverse time and variate dependencies as the errors from one stage can affect the other stage and eventually the overall performance.

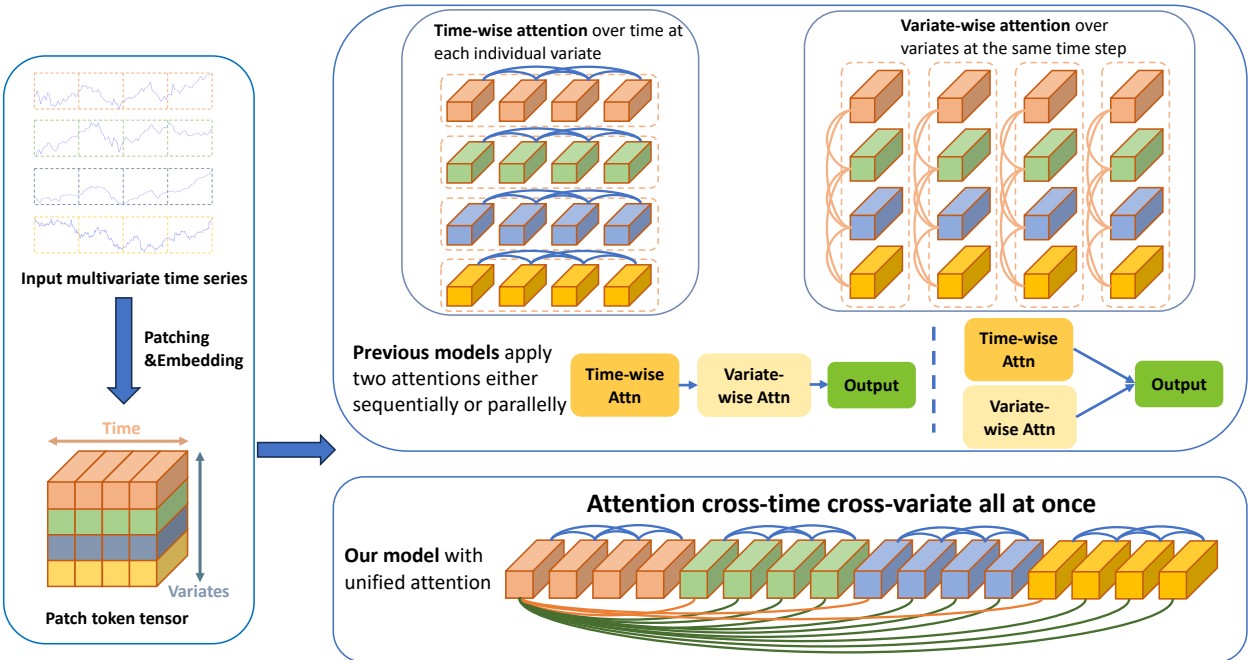

Figure 1: Comparison between our model and previous models. Previous models apply time-wise attention and variate-wise attention modules either sequentially or parallelly, which cannot capture cross-time cross-variate dependencies (i.e., green links) simultaneously like our model.

Additionally, either two parallel or sequential attention mechanisms cannot explicitly model the direct dependencies across different variates and different times, which we show in Figure 1. Regardless of how previous works apply time-wise attention and variate-wise attention parallelly or sequentially, they would still lack the green links to capture cross-time cross-variate dependencies (aka inter-series intra-series dependencies) simultaneously as in our model.

To further explain, as we illustrate in Figure 2, the time series of variate 1 during period 1 shares the same trend with the time series of variate 2 during period 2. This type of correlations cannot be directly modeled by previous works as it requires directly modeling cross-time cross-variate dependencies simultaneously. This type of correlation is important as it generally exists in real-world data as we further demonstrate in Sec 3.

To mitigate the limitations of previous works, in this paper, we revisit the structure of multivariate time series transformers and propose a time series transformer with unified attention (*UniTST*) as a fundamental backbone for multivariate forecasting. Technically, we flatten all patches from different variates into a unified sequence and adopt the attention for inter-variate and intra-variate dependencies simultaneously. To mitigate the high memory cost associated with the flattening strategy, we further develop a dispatcher mechanism to reduce complexity from quadratic to linear. Our contributions are summarized as follows:

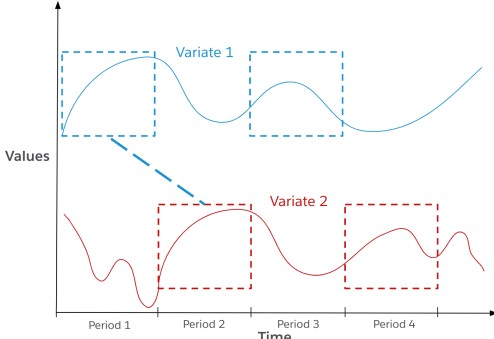

Figure 2: Explicit correlation between two sub-series at different periods from two different variates (i.e., strong correlation between period 1 of variate 1 and period 2 and variate 2).

- We point out the limitation of previous transformer models for multivariate time series forecasting: their lack of ability to simultaneously capture both inter-variate and intra-variate dependencies. With evidence in real-world data, we demonstrate that these dependencies are important and commonly exist.

- To mitigate the limitation, we propose UniTST as a simple, general yet effective transformer for modeling multivariate time series data, which flattens all patches from different variates into a unified sequence to effectively capture inter-variate and intra-variate dependencies.

- Despite the simple designs used in UniTST, we empirically demonstrate that UniTST achieves state-of-the-art performance on real-world benchmarks for both long-term and short-term forecasting with improvements up to 13%. In addition, we provide results of the ablation study and visualization to further demonstrate the effectiveness of our model.

## 2 Related Work

Recently, many Transformer-based models have been also proposed for multivariate time series forecasting and demonstrated great potential (Liu et al., 2021a; Wu et al., 2021; Li et al., 2021; Zhang & Yan, 2023; Zhou et al., 2022; Li et al., 2019; Luo & Wang, 2024; Zhang et al., 2024; Ilbert et al., 2024). Several approaches (Wu et al., 2021; Li et al., 2021; Zhou et al., 2022) embed temporal tokens that contain the multivariate representation of each time step and utilize attention mechanisms to model temporal dependencies. However, due to the vulnerability to the distribution shift, these models with such channel mixing structure are often outperformed by simple linear models (Zeng et al., 2023; Han et al., 2023). Subsequently, PatchTST (Nie et al., 2023) considers channel independence and models temporal dependencies within each channel to make predictions independently. Nonetheless, it ignores the correlation between variates, which may hinder its performance. To model variate dependencies, in the past two years, several works have been proposed (Liu et al., 2024; Zhang & Yan, 2023; Carlini et al., 2023; Han et al., 2024; Yu et al., 2024; Wu et al., 2023). iTransformer (Liu et al., 2024) models channel dependencies by embedding the whole time series of a variate into a token and using "variate-wise attention" without explicitly modeling on temporal dependencies. ElasTST (Zhang et al., 2024) tackles varied-horizon forecasting with a non-autoregressive design and horizon-invariant embeddings while DeformableTST (Luo & Wang, 2024) reduces Transformers' reliance on patching by using deformable attention.

Additionally, several methods proposed different modules to capture both time and variate dependencies. However, they can either sequentially or parallelly capture time and variate dependencies and are not able to capture them simultaneously. In the later Section 3, we show the importance of simultaneously capturing both time and variate dependencies by providing empirical evidence in real-world data. Crossformer (Zhang & Yan, 2023) uses the encoder-decoder architecture with two-stage attention layers to sequentially model cross-time dependencies and then cross-variate dependencies. CARD (Carlini et al., 2023) employs the encoder-only architecture utilizing a similar sequential two-stage attention mechanism for cross-time, cross-channel dependencies and a token blend module to capture multi-scale information. Leddam (Yu et al., 2024) designs a learnable decomposition and a dual attention module that parallelly model inter-variate dependencies with "channel-wise attention" and intra-variate temporal dependencies with "auto-regressive attention". In summary, these works generally model intra-variate and inter-variate dependencies separately (either sequentially or parallelly), and aggregate these two types of information to get the outputs. In contrast, our model has a general ability to directly capture inter-variate and intra-variate dependencies simultaneously, which is more effective. We provide more discussion on the comparison in Section 4.2.

Moreover, CATS (Lu et al., 2024) constructs auxiliary series and capture inter-series dependencies from auxiliary series. In contrast, our method is applied directly on the original series with considering all multivariate as a unified sequence. CrossGNN (Huang et al., 2023), as a GNN-based method, proposes a cross interation layer to capture cross-scale interation on the time dimension and cross-variate interaction on the variate dimension. However, it still relies on a sequential manner to capture cross-time and cross-variate dependencies. Similar to it, TimeXer (Wang et al., 2024) sequentially capture cross-time and cross-variate dependencies by ingesting external information from exogenous variables. With the same goal as our work, LIFT (Zhao & Shen, 2024) also aims to capture cross-time and cross-variate dependencies simultaneously. However, it requires directly calculations on leading indicators for each pair of variates and applies leading indicators lagged variates, which may need massive computational costs. In contrast, our proposed method UniTST can model the cross-time and cross-variate from the time series sequence without explicit calculation on leading indicators.

# 3 Preliminary and Motivation

In multivariate time series forecasting, given historical observations $\mathbf{X}_{:,t:t+L} \in \mathbb{R}^{N \times L}$ with $L$ time steps and $N$ variates, the task is to predict the future $S$ time steps, i.e., $\mathbf{X}_{:,t+L+1:t+L+S} \in \mathbb{R}^{N \times S}$. For convenience, we denote $\mathbf{X}_{i,:} = \mathbf{x}^{(i)}$ as the whole time series of the $i$-th variate and $\mathbf{X}_{:,t}$ as the recorded time points of all variates at time step $t$.

To illustrate the diverse cross-time and cross-variate dependencies from real-world data, we use the following correlation coefficient between $\mathbf{x}_{t:t+L}^{(i)}$ and $\mathbf{x}_{t+L:t+2L}^{(j)}$ to measure it,

**Definition 1** (Cross-Time Cross-Variate Correlation Coefficient)**.**

$$R^{(i,j)}(t, t', L) = \frac{\text{Cov}(\mathbf{x}_{t:t+L}^{(i)}, \mathbf{x}_{t':t'+L}^{(j)})}{\sigma^{(i)}\sigma^{(j)}} = \frac{1}{L}\sum_{k=0}^{L}\frac{\mathbf{x}_{t+k}^{(i)} - \mu^{(i)}}{\sigma^{(i)}} \cdot \frac{\mathbf{x}_{t'+k}^{(j)} - \mu^{(j)}}{\sigma^{(j)}}, \tag{1}$$

where $\mu^{(\cdot)}$ and $\sigma^{(\cdot)}$ are the mean and standard deviation of corresponding time series patches.

Utilizing the above correlation coefficient, we can quantify and further understand the diverse cross-time cross-variate correlation. We visualize the correlation coefficient between different time periods from two different variates in Figure 3. We split the time series into several patches and each patch denotes a time period containing 16 time steps. In Figure 3, we can see that, first, given a pair of variates, the inter-variate dependencies are quite different for different patches. Looking at the column of Patch 20 in variate 10, it is strongly correlated with patch 3, 5, 11, 20, 24 of variate 0, while it is very weakly correlated with all other patches from variate 0. It suggests that there is no consistent correlation pattern for different patch pairs of two variates (i.e., not all the same coefficient at a row/column in the correlation map) and inter-variate dependencies are actually at the fine-grained patch level. Therefore, previous transformer-based models have a deficiency in directly capturing this kind of dependencies. The reason is that they either only capture the dependencies for the whole time series between two variates without considering the fine-grained temporal dependencies across different variates (Liu

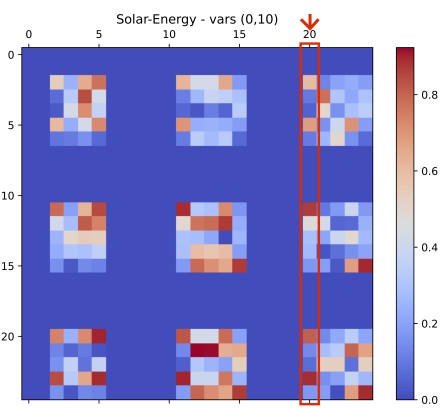

Figure 3: Correlation between patches from different variates. x-axis: patch indices in variate 10, y-axis: patch indices in variate 0.

et al., 2024) or use two separate attention mechanisms (Zhang & Yan, 2023; Carlini et al., 2023; Yu et al., 2024) which are indirect and unable to explicitly learn these dependencies. In Appendix A, we provide more examples to demonstrate the ubiquity and the diversity of these cross-time cross-variate correlations.

Motivated by the deficiency of previous models in capturing these important dependencies, in this work, we aim to propose a model with the ability to explicitly directly capture cross-time cross-variate interactions for multivariate data.

# 4 Methodology

In this section, we describe our proposed Transformer-based method (UniTST) for modeling inter-variate and intra-variate dependencies for multivariate time series forecasting. Then, we discuss and compare our model with previous Transformer-based models in detail.

## 4.1 Model Structure Overview

We illustrate our proposed UniTST with a unified attention mechanism in Figure 4.

**Embedding the patches from different variates as the tokens** Given the time series with $N$ variates $X \in \mathbb{R}^{N \times T}$, we divide each univariate time series $x^i$ into patches as in Nie et al. (2023); Zhang & Yan (2023).

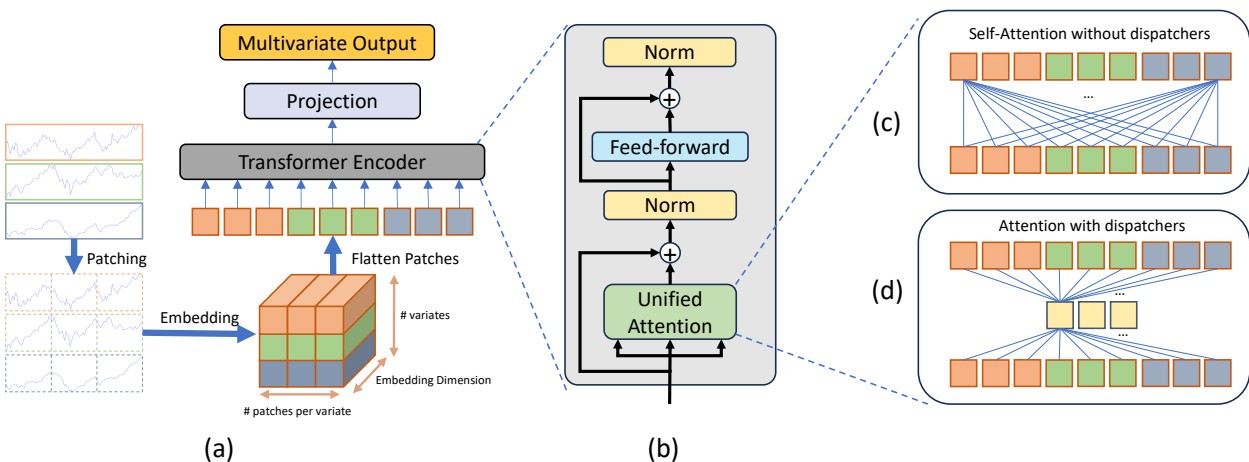

Figure 4: Framework Overview. We flatten the patches from all variates into a sequence as the input of the Transformer Encoder and replace the original self-attention with the proposed unified attention with dispatchers to reduce the memory complexity.

With the patch length $l$ and the stride $s$, for each variate $i$, we obtain a patch sequence $x_p^i \in \mathbb{R}^{p \times l}$ where $p$ is the number of patches. Considering all variates, the tensor containing all patches is denoted as $X_p \in \mathbb{R}^{N \times p \times l}$, where $N$ is the number of variates. With each patch as a token, the 2D token embeddings are generated using a linear projection with position embeddings:

$$H = \text{Embedding}(X_p) = X_p W + W_{pos} \in \mathbb{R}^{N \times p \times d}, \tag{2}$$

where $W \in \mathbb{R}^{l \times d}$ is the learnable projection matrix and $W_{pos} \in \mathbb{R}^{N \times p \times d}$ is the learnable position embeddings. With 2D token embeddings, we denote $H^{(i,k)}$ is the token embedding of the $k$-th patches in the $i$-th variate, resulting in $N \times p$ tokens.

**Self attention on the flattened patch sequence** Considering any two tokens, there are two relationships: 1) they are from the same variate; 2) they are from two different variates. These represent intra-variate and cross-variate dependencies, respectively. A desired model should have the ability to capture both types of dependencies, especially cross-variate dependencies. To capture both intra-variate and cross-variate dependencies among tokens, we flatten the 2D token embedding matrix $H$ into a 1D sequence with $N \times p$ tokens. We use this 1D sequence $X' \in \mathbb{R}^{(N \times p) \times d}$ as the input and feed it to a vanilla Transformer encoder. The multi-head self-attention (MSA) mechanism is directly applied to the 1D sequence:

$$O = \text{MSA}(Q, K, V) = \text{Softmax}(\frac{QK^T}{\sqrt{d_k}})V, \tag{3}$$

with the query matrix $Q = X'W_Q \in \mathbb{R}^{(N \times p) \times d_k}$, the key matrix $K = X'W_K \in \mathbb{R}^{(N \times p) \times d_k}$, the value matrix $V = X'W_V \in \mathbb{R}^{(N \times p) \times d}$, and $W_Q, W_K \in \mathbb{R}^{d \times d_k}$, $W_V \in \mathbb{R}^{d \times d}$. The MSA helps the model to capture dependencies among all tokens, including both intra-variate and cross-variate dependencies. However, the MSA results in an attention map with the memory complexity of $O(N^2 p^2)$, which is very costly when we have a large number of variates $N$.

**Dispatchers** In order to mitigate the complexity of possible large $N$, we further propose a dispatcher mechanism to aggregate and dispatch the dependencies among tokens. We add $k(k << N)$ learnable embeddings as dispatchers and use cross attention to distribute the dependencies. The dispatchers aggregate the information from all tokens by using the dispatcher embeddings $D$ as the query and the token embeddings as the key and value:

$$D' = \text{Attention}(DW_{Q_1}, X'W_{K_1}, X'W_{V_1}) = \text{Softmax}(\frac{DW_{Q_1}(X'W_{K_1})^T}{\sqrt{d_k}})X'W_{V_1}, \tag{4}$$

where the complexity is $O(kNp)$. After that, the dispatchers distribute the dependencies information to all tokens by setting the token embeddings as the key and the dispatcher embeddings as the key and value:

$$O' = \text{Attention}(X'W_{Q_2}, D'W_{K_2}, D'W_{V_2}) = \text{Softmax}(\frac{X'W_{Q_2}(D'W_{K_2})^T}{\sqrt{d_k}})D'W_{V_2}, \quad (5)$$

where the complexity is also $O(kNp)$. Therefore, the overall complexity of our dispatcher mechanism is $O(kNp)$, instead of $O(N^2p^2)$ if we directly use self-attention on the flattened patch sequence. With the dispatcher mechanism, the dependencies between any two patches can be explicitly modeled through attention, no matter if they are from the same variate or different variates.

In a transformer block, the output of attention $O'$ is passed to a BatchNorm Layer and a feedforward layer with residual connections. After stacking several layers, the token representations are generated as $Z^{N \times D}$. In the end, a linear projection is used to generate the prediction $\hat{\mathbf{X}} \in \mathbb{R}^{N \times S}$.

**Loss function**   The Mean-Squared Error (MSE) loss is used as the objective function to measure the difference between the ground truth and the generated predictions: $\mathcal{L} = \frac{1}{NS}\sum_i^N(\hat{\mathbf{X}}^{(i)} - \mathbf{X}_{i,t+L+1:t+S})^2$

## 4.2   Discussion and Comparison with Previous Models

Our proposed model is an encoder-only transformer model containing a unified attention mechanism with dispatchers. The model explicitly learns both intra-variate and inter-variate temporal dependencies among different patch tokens through attention, which means that it can directly capture the correlation between two time series at different periods from different variates. In contrast, these dependencies cannot be directly and explicitly captured by previous works which claim that they model variate dependencies (Liu et al., 2024; Zhang & Yan, 2023; Carlini et al., 2023; Yu et al., 2024).

For example, iTransformer (Liu et al., 2024) captures variate dependencies using the whole time series of a variate as a token. It loses the ability to capture the fine-grained temporal dependencies across channels or within a channel. Crossformer (Zhang & Yan, 2023) and CARD (Carlini et al., 2023) both propose to use a sequential two-stage attention mechanism to first capture dependencies on time dimensions and then capture dependencies on variate dimensions. This sequential manner does not directly capture cross-time cross-variate dependencies simultaneously, which makes them less effective as shown in their empirical performance. In contrast, our proposed model uses a more unified attention on a flattened patch sequence with all patches from different channels, allowing direct and explicit modeling cross-time cross-variate dependencies. In addition, Yu et al. (2024) propose a dual attention module with an iTransformer-like encoder to inter-variate dependencies and an auto-regressive self-attention on each channel to capture intra-variate dependencies separately. In this way, it also cannot directly capture cross-variate temporal dependencies between two patch tokens at different time steps from different variates (e.g., $H^{(i,k)}$, while our model is able to directly capture these dependencies.

Worth noting that our proposed model is a more general case to directly capture intra-variate and inter-variate dependencies at a more fine-grained level (i.e., patch level from different variates at different times). Moreover, our model employs simple architectures that can be easily implemented while the empirical results show the effectiveness of our model in Section 5.1. Additionally, we provide the analysis of computational complexity in Appendix B.

# 5   Experiments

We conduct comprehensive experiments to evaluate our proposed model UniTST and compare it with 11 representative baselines for both short-term and long-term time series forecasting on 13 datasets. Additionally, we further dive deeper into model analysis to examine the effectiveness of our model from different aspects.

### 5.1 Forecasting Results

We conduct extensive experiments to compare our model with several representative time series models for both short-term and long-term time series forecasting.

**Experimental Setting** We conduct all the experiments with PyTorch (Paszke et al., 2019) and utilize a single NVIDIA A100 GPU with 40GB memory. We describe the hyperparameter choices used in our experiments in the following. For the optimizer, we use ADAM (Kingma & Ba, 2015) with the learning rate in $\{10^{-3}, 5 \times 10^{-4}, 10^{-4}\}$. The batch sizes are selected from $\{16, 32, 64, 128\}$ depending on the dataset sizes. The maximum number of training epochs is set to 100 as in Nie et al. (2023). Meanwhile, we also use the early stop strategy to stop the training when the loss does not decrease in 10 epochs. The number of layers of our Transformer blocks is selected from $\{2,3,4\}$. The hidden dimension of $D$ is set from $\{128, 256, 512\}$.

For the experimental results of our model, we report the averaged results with 5 runs with different seeds. For the results of previous models, we reuse the results from iTransformer paper (Liu et al., 2024) as we are using the same experimental setting.

**Baselines** We select 11 well-known forecasting models as our baselines, including (1) Transformer-based models: iTransformer (Liu et al., 2024), Crossformer (Zhang & Yan, 2023), FEDformer (Zhou et al., 2022), Stationary (Liu et al., 2022b), PatchTST (Nie et al., 2023); (2) Linear-based methods: DLinear (Zeng et al., 2023), RLinear (Li et al., 2023), TiDE (Das et al., 2023); (3) Temporal Convolutional Network (TCN)-based methods: TimesNet (Wu et al., 2023), SCINet (Liu et al., 2022a); and (4) MLP-based method: SOFTS (Han et al., 2024).

Table 1: Multivariate long-term forecasting results with prediction lengths $S \in \{96, 192, 336, 720\}$ and fixed lookback length $T = 96$. Results are averaged from all prediction lengths. Full results are listed in Appendix C.3.1, Table 6.

| Models | UniTST (Ours) | | SOFTS (2024) | | iTransformer (2024) | | RLinear (2023) | | PatchTST (2023) | | Crossformer (2023) | | TiDE (2023) | | TimesNet (2023) | | DLinear (2023) | | SCINet (2022a) | | FEDformer (2022) | | Stationary (2022b) | |
|---|---|---|---|---|---|---|---|---|---|---|---|---|---|---|---|---|---|---|---|---|---|---|---|---|
| Metric | MSE | MAE | MSE | MAE | MSE | MAE | MSE | MAE | MSE | MAE | MSE | MAE | MSE | MAE | MSE | MAE | MSE | MAE | MSE | MAE | MSE | MAE | MSE | MAE |
| ECL | **0.166** | **0.262** | 0.174 | 0.264 | 0.178 | 0.270 | 0.219 | 0.298 | 0.205 | 0.290 | 0.244 | 0.334 | 0.251 | 0.344 | 0.192 | 0.295 | 0.212 | 0.300 | 0.268 | 0.365 | 0.214 | 0.327 | 0.193 | 0.296 |
| ETTm1 | **0.379** | **0.394** | 0.393 | 0.403 | 0.407 | 0.410 | 0.414 | 0.407 | 0.387 | 0.400 | 0.513 | 0.496 | 0.419 | 0.419 | 0.400 | 0.406 | 0.403 | 0.407 | 0.485 | 0.481 | 0.448 | 0.452 | 0.481 | 0.456 |
| ETTm2 | **0.280** | **0.326** | 0.287 | 0.330 | 0.288 | 0.332 | 0.286 | 0.327 | 0.281 | 0.326 | 0.757 | 0.610 | 0.358 | 0.404 | 0.291 | 0.333 | 0.350 | 0.401 | 0.571 | 0.537 | 0.305 | 0.349 | 0.306 | 0.347 |
| ETTh1 | 0.442 | 0.435 | 0.449 | 0.442 | 0.454 | 0.447 | 0.446 | **0.434** | 0.469 | 0.454 | 0.529 | 0.522 | 0.541 | 0.507 | 0.458 | 0.450 | 0.456 | 0.452 | 0.747 | 0.647 | **0.440** | 0.460 | 0.570 | 0.537 |
| ETTh2 | **0.363** | **0.393** | 0.373 | 0.400 | 0.383 | 0.407 | 0.374 | 0.398 | 0.387 | 0.407 | 0.942 | 0.684 | 0.611 | 0.550 | 0.414 | 0.427 | 0.559 | 0.515 | 0.954 | 0.723 | 0.437 | 0.449 | 0.526 | 0.516 |
| Exchange | **0.351** | **0.398** | — | — | 0.360 | 0.403 | 0.378 | 0.417 | 0.367 | 0.404 | 0.940 | 0.707 | 0.370 | 0.413 | 0.416 | 0.443 | 0.354 | 0.414 | 0.750 | 0.626 | 0.519 | 0.429 | 0.461 | 0.454 |
| Traffic | 0.439 | 0.274 | **0.409** | **0.267** | 0.428 | 0.282 | 0.626 | 0.378 | 0.481 | 0.304 | 0.550 | 0.304 | 0.760 | 0.473 | 0.620 | 0.336 | 0.625 | 0.383 | 0.804 | 0.509 | 0.610 | 0.376 | 0.624 | 0.340 |
| Weather | **0.242** | **0.271** | 0.255 | 0.278 | 0.258 | 0.278 | 0.272 | 0.291 | 0.259 | 0.281 | 0.259 | 0.315 | 0.271 | 0.320 | 0.259 | 0.287 | 0.265 | 0.317 | 0.292 | 0.363 | 0.309 | 0.360 | 0.288 | 0.314 |
| Solar-Energy | **0.225** | 0.260 | 0.229 | **0.256** | 0.233 | 0.262 | 0.369 | 0.356 | 0.270 | 0.307 | 0.641 | 0.639 | 0.347 | 0.417 | 0.301 | 0.319 | 0.330 | 0.401 | 0.282 | 0.375 | 0.291 | 0.381 | 0.261 | 0.381 |
| 1st Count | **7** | **6** | 1 | 2 | 0 | 0 | 0 | 1 | 0 | 1 | 0 | 1 | 0 | 0 | 0 | 0 | 0 | 0 | 0 | 0 | 1 | 0 | 0 | 0 |

**Long-term forecasting** Following iTransformer (Liu et al., 2024), we use 4 different prediction lengths (i.e., {96, 192, 336, 720}) and fix the lookback window length as 96 for the long-term forecasting task. We evaluate models with MSE (Mean Squared Error) and MAE (Mean Absolute Error) – the lower values indicate better prediction performance. We summarize the long-term forecasting results in Table 1 with the best in **red** and the second underlined. Overall, we can see that UniTST achieves the best results compared with 11 baselines on 7 out of 9 datasets for MSE and 8 out of 9 datasets for MAE. Particularly, iTransformer, as the previous state-of-the-art model, performs worse than our model in most cases of ETT datasets and ECL dataset (which are both from electricity domain). This may indicate that only model multivariate correlation without considering temporal correlation is not effective for some datasets. Meanwhile, the results of PatchTST are also deficient, suggesting that only capturing temporal relationships within a channel is not sufficient as well. In contrast, our proposed model UniTST can better capture temporal relationships both within a variate and across different variates, which leads to better prediction performance. Besides, although Crossformer is claimed to capture cross-time and cross-variate dependencies, it still performs much worse compared with our

approach. The reason is that their sequential design with two attention modules cannot simultaneously and effectively capture cross-time and cross-variate dependencies, while our approach can explicitly model these dependencies at the same time.

Table 2: Full results of the PEMS forecasting task. We compare extensive competitive models under different prediction lengths following the setting of SCINet (2022a). The input length is set to 96 for all baselines. *Avg* means the average results from all four prediction lengths.

| Models | | UniTST (Ours) | | SOFTS (2024) | | iTransformer (2024) | | RLinear (2023) | | PatchTST (2023) | | Crossformer (2023) | | TiDE (2023) | | TimesNet (2023) | | DLinear (2023) | | SCINet (2022a) | | FEDformer (2022) | | Stationary (2022b) | |
|---|---|---|---|---|---|---|---|---|---|---|---|---|---|---|---|---|---|---|---|---|---|---|---|---|---|
| Metric | | MSE | MAE | MSE | MAE | MSE | MAE | MSE | MAE | MSE | MAE | MSE | MAE | MSE | MAE | MSE | MAE | MSE | MAE | MSE | MAE | MSE | MAE | MSE | MAE |
| PEMS03 | 12 | **0.059** | **0.160** | 0.064 | 0.165 | 0.071 | 0.174 | 0.126 | 0.236 | 0.099 | 0.216 | 0.090 | 0.203 | 0.178 | 0.305 | 0.085 | 0.192 | 0.122 | 0.243 | 0.066 | 0.172 | 0.126 | 0.251 | 0.081 | 0.188 |
| | 24 | **0.074** | **0.180** | 0.083 | 0.188 | 0.093 | 0.201 | 0.246 | 0.334 | 0.142 | 0.259 | 0.121 | 0.240 | 0.257 | 0.371 | 0.118 | 0.223 | 0.201 | 0.317 | 0.085 | 0.198 | 0.149 | 0.275 | 0.105 | 0.214 |
| | 48 | **0.104** | **0.213** | 0.114 | 0.223 | 0.125 | 0.236 | 0.551 | 0.529 | 0.211 | 0.319 | 0.202 | 0.317 | 0.379 | 0.463 | 0.155 | 0.260 | 0.333 | 0.425 | 0.127 | 0.238 | 0.227 | 0.348 | 0.154 | 0.257 |
| | 96 | **0.151** | **0.261** | 0.156 | 0.264 | 0.164 | 0.275 | 1.057 | 0.787 | 0.269 | 0.370 | 0.262 | 0.367 | 0.490 | 0.539 | 0.228 | 0.317 | 0.457 | 0.515 | 0.178 | 0.287 | 0.348 | 0.434 | 0.247 | 0.336 |
| | Avg | **0.097** | **0.204** | 0.104 | 0.210 | 0.113 | 0.221 | 0.495 | 0.472 | 0.180 | 0.291 | 0.169 | 0.281 | 0.326 | 0.419 | 0.147 | 0.248 | 0.278 | 0.375 | 0.114 | 0.224 | 0.213 | 0.327 | 0.147 | 0.249 |
| PEMS04 | 12 | **0.070** | **0.172** | 0.074 | 0.176 | 0.078 | 0.183 | 0.138 | 0.252 | 0.105 | 0.224 | 0.098 | 0.218 | 0.219 | 0.340 | 0.087 | 0.195 | 0.148 | 0.272 | 0.073 | 0.177 | 0.138 | 0.262 | 0.088 | 0.196 |
| | 24 | **0.082** | **0.189** | 0.088 | 0.194 | 0.095 | 0.205 | 0.258 | 0.348 | 0.153 | 0.275 | 0.131 | 0.256 | 0.292 | 0.398 | 0.103 | 0.215 | 0.224 | 0.340 | 0.084 | 0.193 | 0.177 | 0.293 | 0.104 | 0.216 |
| | 48 | 0.104 | 0.216 | 0.110 | 0.219 | 0.120 | 0.233 | 0.572 | 0.544 | 0.229 | 0.339 | 0.205 | 0.326 | 0.409 | 0.478 | 0.136 | 0.250 | 0.355 | 0.437 | **0.099** | **0.211** | 0.270 | 0.368 | 0.137 | 0.251 |
| | 96 | 0.137 | 0.256 | 0.135 | 0.244 | 0.150 | 0.262 | 1.137 | 0.820 | 0.291 | 0.389 | 0.402 | 0.457 | 0.492 | 0.532 | 0.190 | 0.303 | 0.452 | 0.504 | **0.114** | **0.227** | 0.341 | 0.427 | 0.186 | 0.297 |
| | Avg | 0.098 | 0.208 | 0.102 | 0.208 | 0.111 | 0.221 | 0.526 | 0.491 | 0.195 | 0.307 | 0.209 | 0.314 | 0.353 | 0.437 | 0.129 | 0.241 | 0.295 | 0.388 | **0.092** | **0.202** | 0.231 | 0.337 | 0.127 | 0.240 |
| PEMS07 | 12 | **0.057** | 0.153 | **0.057** | **0.152** | 0.067 | 0.165 | 0.118 | 0.235 | 0.095 | 0.207 | 0.094 | 0.200 | 0.173 | 0.304 | 0.082 | 0.181 | 0.115 | 0.242 | 0.068 | 0.171 | 0.109 | 0.225 | 0.083 | 0.185 |
| | 24 | 0.075 | 0.174 | **0.073** | **0.173** | 0.088 | 0.190 | 0.242 | 0.341 | 0.150 | 0.262 | 0.139 | 0.247 | 0.271 | 0.383 | 0.101 | 0.204 | 0.210 | 0.329 | 0.119 | 0.225 | 0.125 | 0.244 | 0.102 | 0.207 |
| | 48 | 0.107 | 0.208 | **0.096** | **0.195** | 0.110 | 0.215 | 0.562 | 0.541 | 0.253 | 0.340 | 0.311 | 0.369 | 0.446 | 0.495 | 0.134 | 0.238 | 0.398 | 0.458 | 0.149 | 0.237 | 0.165 | 0.288 | 0.136 | 0.240 |
| | 96 | 0.133 | 0.228 | **0.120** | **0.218** | 0.139 | 0.245 | 1.096 | 0.795 | 0.346 | 0.404 | 0.396 | 0.442 | 0.628 | 0.577 | 0.181 | 0.279 | 0.594 | 0.553 | 0.141 | 0.234 | 0.262 | 0.376 | 0.187 | 0.287 |
| | Avg | 0.093 | 0.191 | **0.087** | **0.184** | 0.101 | 0.204 | 0.504 | 0.478 | 0.211 | 0.303 | 0.235 | 0.315 | 0.380 | 0.440 | 0.124 | 0.225 | 0.329 | 0.395 | 0.119 | 0.234 | 0.165 | 0.283 | 0.127 | 0.230 |
| PEMS08 | 12 | **0.073** | 0.174 | 0.074 | **0.171** | 0.079 | 0.182 | 0.133 | 0.247 | 0.168 | 0.232 | 0.165 | 0.214 | 0.227 | 0.343 | 0.112 | 0.212 | 0.154 | 0.276 | 0.087 | 0.184 | 0.173 | 0.273 | 0.109 | 0.207 |
| | 24 | **0.096** | **0.197** | 0.104 | 0.201 | 0.115 | 0.219 | 0.249 | 0.343 | 0.224 | 0.281 | 0.215 | 0.260 | 0.318 | 0.409 | 0.141 | 0.238 | 0.248 | 0.353 | 0.122 | 0.221 | 0.210 | 0.301 | 0.140 | 0.236 |
| | 48 | **0.141** | **0.239** | 0.164 | 0.253 | 0.186 | 0.235 | 0.569 | 0.544 | 0.321 | 0.354 | 0.315 | 0.355 | 0.497 | 0.510 | 0.198 | 0.283 | 0.440 | 0.470 | 0.189 | 0.270 | 0.320 | 0.394 | 0.211 | 0.294 |
| | 96 | **0.210** | 0.275 | 0.211 | **0.253** | 0.221 | 0.267 | 1.166 | 0.814 | 0.408 | 0.417 | 0.377 | 0.397 | 0.721 | 0.592 | 0.320 | 0.351 | 0.674 | 0.565 | 0.236 | 0.300 | 0.442 | 0.465 | 0.345 | 0.367 |
| | Avg | **0.130** | 0.221 | 0.138 | **0.219** | 0.150 | 0.226 | 0.529 | 0.487 | 0.280 | 0.321 | 0.268 | 0.307 | 0.441 | 0.464 | 0.193 | 0.271 | 0.379 | 0.416 | 0.158 | 0.244 | 0.286 | 0.358 | 0.201 | 0.276 |
| 1st Count | | **11** | **8** | 4 | 6 | 0 | 0 | 0 | 0 | 0 | 0 | 0 | 0 | 0 | 0 | 0 | 0 | 0 | 0 | 2 | 2 | 0 | 0 | 0 | 0 |

**Short-term forecasting** Besides long-term forecasting, we also conduct experiments for short-term forecasting with 4 prediction lengths (i.e., {12, 24, 48, 96}) on PEMS datasets as in SCINet (Liu et al., 2022a) and iTransformer (Liu et al., 2024). Full results on 4 PEMS datasets with 4 different prediction lengths are shown in Table 2. Generally, our model outperforms other baselines on all prediction lengths and all PEMS datasets, which demonstrates the superiority of capturing cross-channel cross-time relationships for short-term forecasting. Additionally, we observe that PatchTST usually underperforms iTransformer by a large margin, suggesting that modeling channel dependencies is necessary for PEMS datasets. The worse results of iTransformer, compared with our model, indicate that cross-channel temporal relationships are important and should be captured on these datasets.

## 5.2 Model Analysis

**Ablation study** We conduct the ablation study to verify the effectiveness of our dispatcher module by using the same setting (e.g., the number of layers, hidden dimensions, batch size) for comparing the our model with and without dispatchers. In Table 3, we can see that adding dispatchers helps to reduce GPU usage. In

Table 3: The effectiveness of our dispatcher module. OOM indicates the "Out of Memory" error on GPUs (we a single A100 GPU of memory 40GB).

| | ETTm1 | | Weather | | ECL | | Traffic | |
|---|---|---|---|---|---|---|---|---|
| | MSE | Mem | MSE | Mem | MSE | Mem | MSE | Mem |
| w/o dispatchers | 0.385 | 2.56GB | 0.247 | 9.17GB | OOM | OOM | OOM | OOM |
| w/ dispatchers | 0.379 | 2.33GB | 0.242 | 5.13GB | 0.166 | 13.32GB | 0.439 | 22.87GB |

ECL and Traffic, the version without dispatchers even leads to out-of-memory (OOM) issues. Moreover, we observe that the memory reduction becomes more significant when the number of variates increases. On ETTm1 with 7 variates, the memory only reduces from 2.56GB to 2.33GB, while on ECL and Traffic, it reduces from OOM (more than 40GB) to 13.32GB and 22.87GB, respectively.

**The effect of different lookback lengths** We also investigate how different lookback lengths would change the forecasting performance. With increased lookback lengths, we compare the forecasting performance of our model with that of several representative baselines in Figure 5. The results show that, when using a relatively short lookback length (i.e., 48), our model generally outperforms other models by a large margin. It suggests that our model has a more powerful learning ability to capture the dependencies even with a short lookback length, while other models usually require longer lookback lengths to provide good performance. Moreover, by increasing the lookback length, the performances of our model and PatchTST usually improve, whereas the performance of Transformer remains almost the same on ECL dataset.

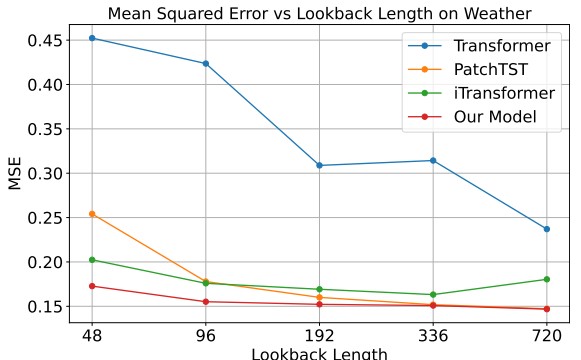 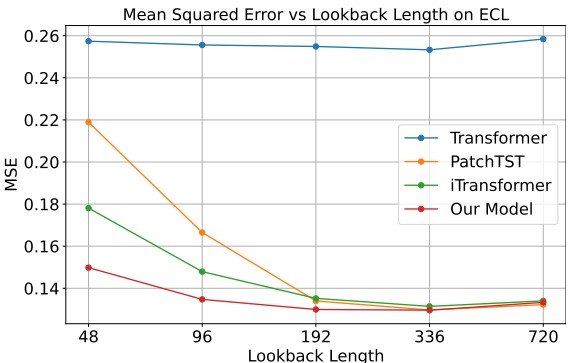

Figure 5: Performance with different lookback lengths and fixed prediction length $S = 96$.

**The effect of different patch sizes** As we use patching in our model, we further examine the effect of different patch sizes. The patch size and the lookback length together determine the number of tokens for a variate. In Figure 6, we demonstrate the performance by varying different patch sizes and lookback lengths. With lookback length of 64, the performance of using patch size 64 is much worse than that of patch size 8 It indicates that, when the number of tokens of a variate is extremely small (i.e., only 1 token for lookback length 64), the performance is not satisfactory as no enough fine-grained information. This could also be the reason why iTransformer may be not ideal in some cases - it use exactly a single token for a variate. Additionally, we observe that, for different lookback lengths, too small or too large patch size can lead to bad performance. The reason may be that too many tokens or too less tokens would increase the difficulty of training.

**The number of dispatchers** In our model, we propose to use several dispatchers to reduce the memory complexity with the number of dispatchers as a hyper-parameter. Here, we dive deep into the tradeoff between GPU memory and MSE by varying the number of dispatchers. In Table 4, we demonstrate the performance and GPU memory of different numbers of dispatchers on Weather and ECL with the prediction length as 96. The results show that, with only 5 dispatchers, the performance is usually worse than with more dispatchers. It suggests that we should avoid using too few dispatchers as it may affect the model performance. However, with fewer dispatchers, the GPU memory usage is less as shown in our complexity analysis in Section 4.1. For larger datasets like ECL, increasing the number of dispatchers leads to more signifi-

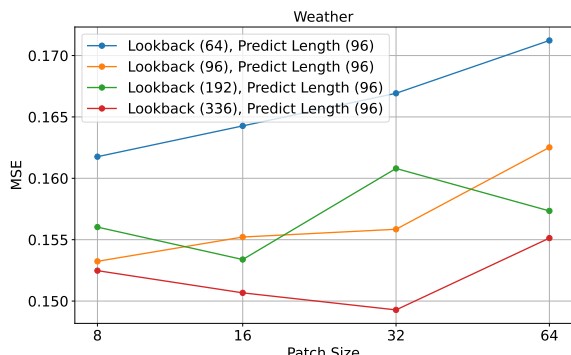

Figure 6: Performance with different patch sizes and lookback length.

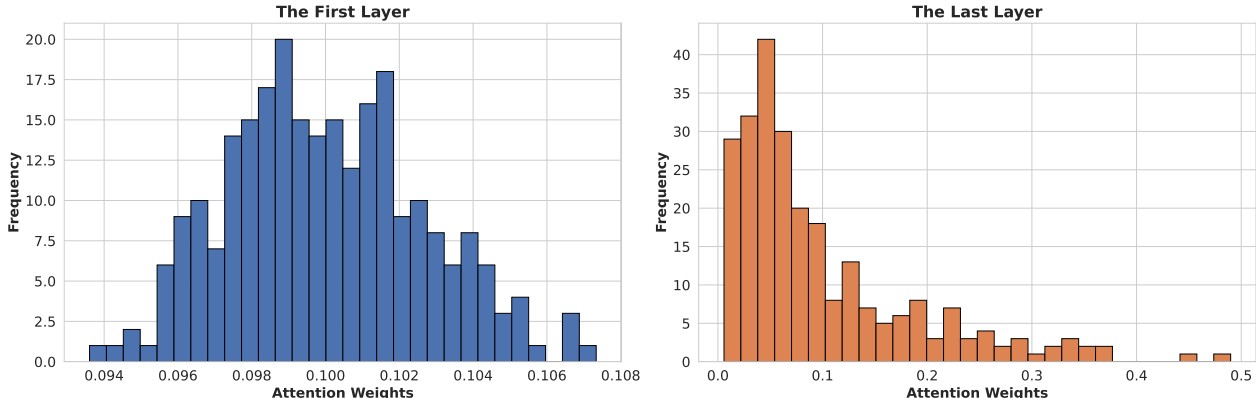

Figure 7: The distributions of multiplied attention weights between two patch tokens on Weather.

cant memory increase, compared with the smaller dataset (i.e., Weather).

Table 4: The performance and GPU memory usage of varying dispatchers on Weather and ECL.

| The number of dispatchers | | 5 | 10 | 20 | 50 |
|---|---|---|---|---|---|
| Weather | MSE | 0.1575 | 0.1552 | 0.1573 | 0.1566 |
| | GPU Memory (GB) | 2.165 | 2.191 | 2.233 | 2.405 |
| ECL | MSE | 0.1348 | 0.1347 | 0.1343 | 0.1338 |
| | GPU Memory (GB) | 12.807 | 13.389 | 14.335 | 16.509 |

**Attention Weights** With our dispatcher module, we have two attention weights matrices, one from patch tokens to dispatchers and one from dispatchers to patch tokens, with the size $N \times k$ and $k \times N$, respectively. Multiplying these two attention matrices gives us a new multiplied attention matrix with the size $N \times N$ that directly indicates the importance between two patch tokens. We demonstrate the multiplied attention weights from the first layer and the last layer in Figure 7. As shown, in the last layer, the distribution is visibly shifted to the left side, meaning that most of the token pairs have low attention weights, while a few token pairs have high attention weights. It may suggest that the last layer indeed learns how to distribute the information to important tokens. In contrast, the first layer has a more even distribution of attention weights, indicating that it distributes information more evenly to all tokens.

**The importance of cross-variate cross-time dependencies** With the multiplied attention weights, we further demonstrate the percentages of patch token pairs from different variables and different times for groups of patch tokens pairs with varied attention weights in Figure 8. We observe that the groups of patch token pairs with higher attention weights have a higher percentage of pairs from different variates and different times. For example, for all token pairs, the percentage is 87.50, while the percentage is 89.91 for top 0.5% token pairs with the highest attention weights. It suggests that more pairs of patch tokens with high attention weights come from different variates and times. Therefore, effectively modeling cross-variate cross-time is crucial for multivariate time series forecasting.

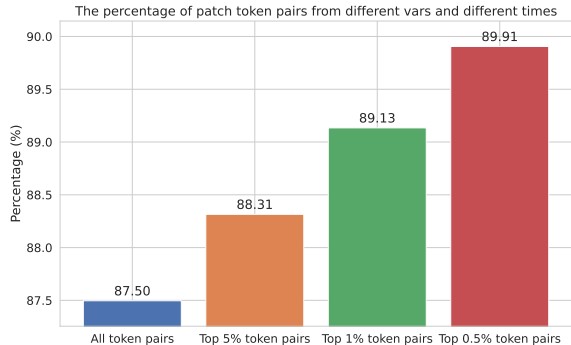

Figure 8: Patch token pairs with higher top attention weights are more likely from different variates and different times.

# 6  Case Studies

## 6.1  Visualization of Multivariate Correlations

To further investigate the ability of capturing multivariate correlations, in Figure 9, we provide two cases of visualizations on the correlation map of multivariate relationships in the predicted time series from Solar-Energy. We can find that, the correlation map of UniTST is similar to the correlation map of ground truth time series, which indicates that the variate dependencies are well-captured by UniTST. In contrast, compared with ours, the correlation map of iTransformer is less aligned with that of ground truth time series.

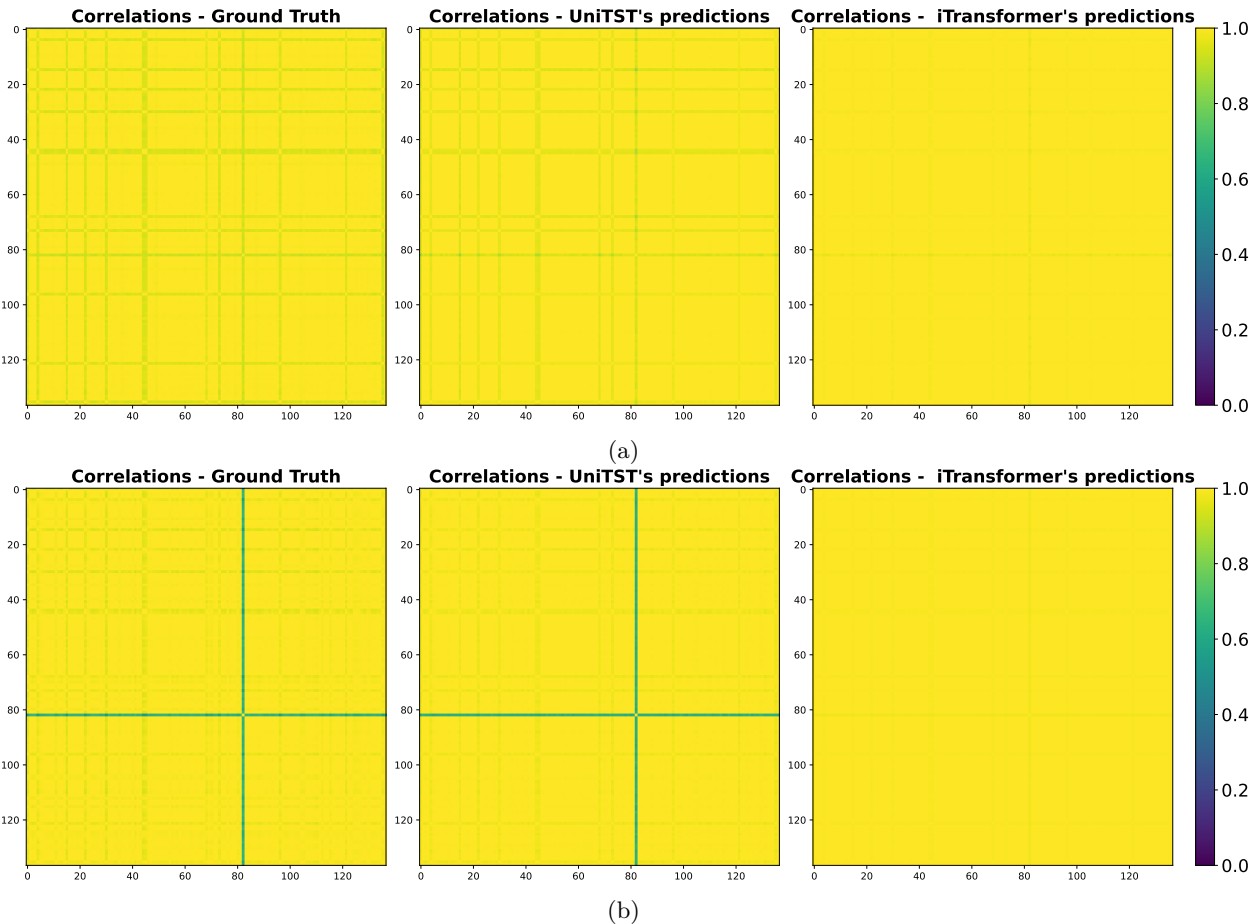

Figure 9: The correlation maps of multivariate relationship with different models. The x-axis and y-axis both represent the variate index. The darker color indicates the stronger correlation.

# 7  Conclusion

In this work, we first point out the limitation of previous works on time series transformers for multivariate forecasting: their lack of ability to effectively capture inter-series and intra-series dependencies simultaneously. We further demonstrate that inter-series and intra-series dependencies are crucial for multivariate time series forecasting as they commonly exist in real-world data. To mitigate this limitation of previous works, we propose a simple yet effective transformer model UniTST with a dispatcher mechanism to effectively capture inter-series and intra-series dependencies. The experiments on 13 datasets for time series forecasting show that our model achieves superior performance compared with many representative baselines.

Moreover, we conduct the ablation study to verify the effectiveness of our dispatcher mechanism and demonstrate the importance of inter-series and intra-series dependencies. Lastly, we also provide model analyses to demonstrate the importance of inter-series and intra-series dependencies, and specifically, a case study on visualization of multivariate correlations validates the ability of our model to capture variate (inter-series) dependencies. Our study emphasizes the necessity and effectiveness of simultaneously capturing inter-variate and intra-variate dependencies in multivariate time series forecasting, and our proposed designs may represent a step toward this goal.

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

## A   Diverse Cross-Time and Cross-Variate Dependencies

We further illustrate the cross-time cross-variate correlations on Exchange, Weather, ECL datasets in Figure 10. We can see that correlation patterns for different datasets are quite different. Additionally, even for a specific dataset with different variate pairs, the correlations of cross-variate patch pairs are also very diverse. For example, for Exchange, with variate pairs (1,3), the patches at the same time step are usually strongly correlated. In contrast, with variate pairs (3,4), the patches can sometimes even have zero correlation coefficient. Moreover, in Figure 10, for a specific dataset with a specific pair of variates (i.e., in a subfigure), we have similar observations as we discussed in Sec 3 that there is no consistent correlation pattern for different patch pairs of two variates and inter-variate dependencies are at the fine-grained patch level. These examples further demonstrate the ubiquity and the diversity of these cross-time cross-variate correlations in real data. This also justifies the motivation of this paper – propose a better method to explicitly model cross-time and cross-variate (intra-variate and inter-variate) dependencies.

## B   Discussion on Computational Complexity

Moreover, we provide the computational complexity analysis of different models. As Feedforward networks in different models have similar complexities, we mainly analyze the computational complexity of the attention mechanism. For UniTST, the designed attention mechanism uses cross-attention with dispatchers to reduce the complexity. It results in the complexity as $O(kNp)$ where $k$ is the number of dispatchers, $N$ is the number of variates, and $p$ is the number of patches within a variate. For iTransformer, it utilizes self-attention on the variate dimension, which leads to the complexity as $O(N^2)$. Additionally, PatchTST uses self-attention on the time dimension and treats each variate independently. As a result, the complexity is $O(Np^2)$. We can see that different models have different advantages in different scenarios. For example, when handling data with a long time series but with fewer variates, iTransformer should be faster than others as it doesn't depend on $p$. Comparing UniTST and PatchTST, when $p$ is relatively small, then the complexity should be similar (we set $k$ as 10 in our experiments). UniTST may be slower than iTransformer when the length of the time series and the number of variates are both extremely large. For this extreme scenario, we leave further investigation for future work.

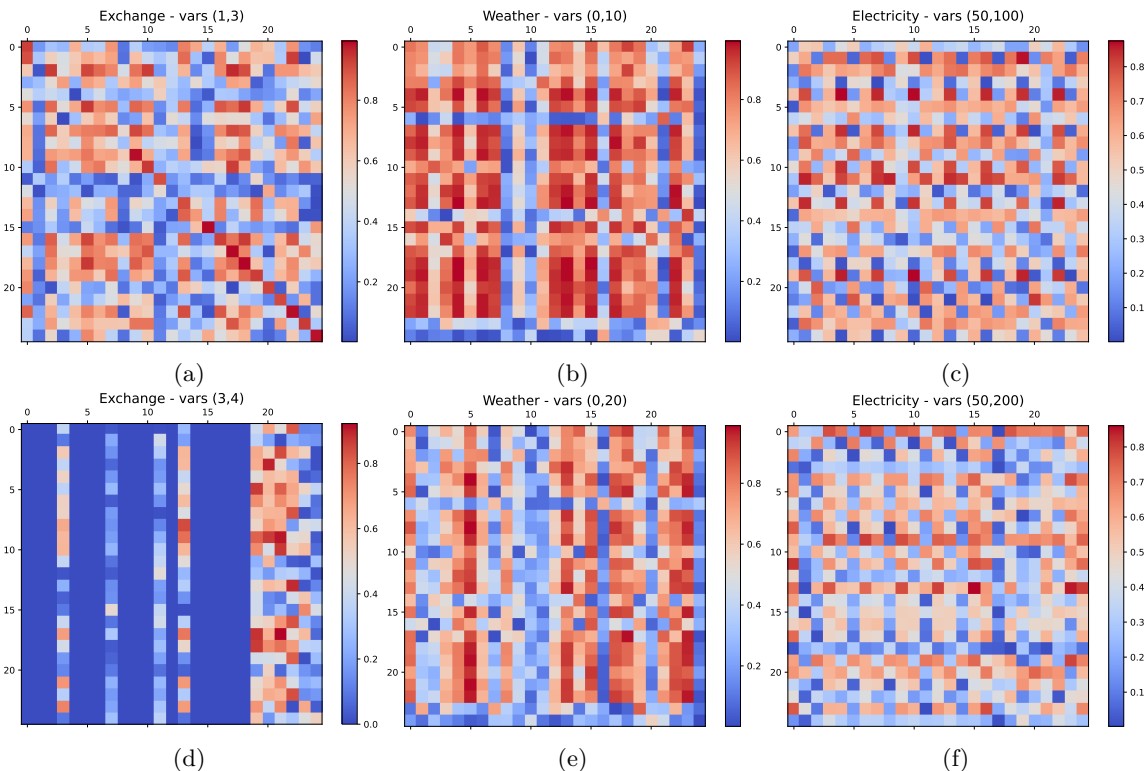

Figure 10: Diverse cross-time cross-variate dependencies commonly exist in real-world data.

# C More on Experiments

## C.1 Datasets

Following Liu et al. (2024), we conduct experiments on 13 real-world datasets to evaluate the performance of our model including (1) a group of datasets – ETT (Li et al., 2021) contains 7 factors of electricity transformer from July 2016 to July 2018. There are four datasets where ETTm1 and ETTm2 are recorded every 15 minutes, and ETTh1 and ETTh2 are recorded every hour; (2) Exchange (Wu et al., 2021) contains daily exchange rates from 8 countries from 1990 to 2016. (3) Weather (Wu et al., 2021) collects the every 10-min data of 21 meteorological factors from the Weather Station of the Max Planck Biogeochemistry Institute in 2020. (4) ECL (Wu et al., 2021) records the electricity consumption data from 321 clients every hour. (5) Traffic (Wu et al., 2021) collects hourly road occupancy rates measured by 862 sensors of San Francisco Bay area freeways from January 2015 to December 2016. (6) Solar-Energy (Lai et al., 2018) records the solar power production of 137 PV plants in 2006, which are sampled every 10 minutes. (7) a group of datasets – PEMS records the public traffic network data in California and collected by 5-minute windows. We use the same four public datasets (PEMS03, PEMS04, PEMS07, PEMS08) adopted in SCINet (Liu et al., 2022a) and iTransformer (Liu et al., 2024). We provide the detailed dataset statistics and descriptions in Table 5.

We also use the same train-validation-test splits as in TimesNet (Wu et al., 2023) and iTransformer (Liu et al., 2024). For the forecasting setting, following iTansformer (Liu et al., 2024), we use the fixed lookback length as 96 in all datasets. In terms of the prediction lengths, we use the varied prediction lengths in {96, 192, 336, 720} for ETT, Exchange, Weather, ECL, Traffic, Solar-Energy. For PEMS datasets, we use the prediction lengths as {12, 24, 48, 96} for short-term forecasting.

Table 5: Detailed dataset statistics. *# variates* denotes the variate number of each dataset. *Dataset Size* denotes the total number of time points in (Train, Validation, Test) split respectively. *Frequency* indicates the sampling interval of data points.

| Dataset Name | # variates | Prediction Length | Dataset Size | Frequency | Information |
|---|---|---|---|---|---|
| ETTh1, ETTh2 | 7 | {96, 192, 336, 720} | (8545, 2881, 2881) | Hourly | Electricity |
| ETTm1, ETTm2 | 7 | {96, 192, 336, 720} | (34465, 11521, 11521) | 15min | Electricity |
| Exchange | 8 | {96, 192, 336, 720} | (5120, 665, 1422) | Daily | Economy |
| Weather | 21 | {96, 192, 336, 720} | (36792, 5271, 10540) | 10min | Weather |
| ECL | 321 | {96, 192, 336, 720} | (18317, 2633, 5261) | Hourly | Electricity |
| Traffic | 862 | {96, 192, 336, 720} | (12185, 1757, 3509) | Hourly | Transportation |
| Solar-Energy | 137 | {96, 192, 336, 720} | (36601, 5161, 10417) | 10min | Energy |
| PEMS03 | 358 | {12, 24, 48, 96} | (15617, 5135, 5135) | 5min | Transportation |
| PEMS04 | 307 | {12, 24, 48, 96} | (10172, 3375, 3375) | 5min | Transportation |
| PEMS07 | 883 | {12, 24, 48, 96} | (16911, 5622, 5622) | 5min | Transportation |
| PEMS08 | 170 | {12, 24, 48, 96} | (10690, 3548, 3548) | 5min | Transportation |

## C.2 Experimental Setting

We conduct all the experiments with PyTorch (Paszke et al., 2019) and utilize a single NVIDIA A100 GPU with 40GB memory. We describe the hyperparameter choices used in our experiments in the following. For the optimizer, we use ADAM (Kingma & Ba, 2015) with the learning rate in $\{10^{-3}, 5 \times 10^{-4}, 10^{-4}\}$. The batch sizes are selected from $\{16, 32, 64, 128\}$ depending on the dataset sizes. The maximum number of training epochs is set to 100 as in Nie et al. (2023). Meanwhile, we also use the early stop strategy to stop the training when the loss does not decrease in 10 epochs. The number of layers of our Transformer blocks is selected from $\{2,3,4\}$. The hidden dimension of $D$ is set from $\{128, 256, 512\}$.

For the experimental results of our model, we report the averaged results with 5 runs with different seeds. For the results of previous models, we reuse the results from iTransformer paper (Liu et al., 2024) as we are using the same experimental setting.

## C.3 Additional Experimental Results

### C.3.1 Full Result of Forecasing

Due to the space limitation, we only display the averaged results over 4 prediction lengths for datasets on long-term forecasting. Here, we provide the full results of long-term forecasting in Table 6. In summary, our model achieves the best results on 24 and 26 out of 36 settings with different prediction lengths among other baselines.

### C.3.2 The correlation maps of multivariate relationship on different datasets

In this section, we provide additional visualizations on the correlationship map of multivariate relationship for ETTm1 dataset in Figure 11. Similar to Figure 9 for Solar-Energy, it also shows that the variate dependencies are well-captured by UniTST.

Table 6: Full results of the long-term forecasting task. We compare extensive competitive models under different prediction lengths following the setting of TimesNet (2023). The input sequence length is set to 96 for all baselines. *Avg* means the average results from all four prediction lengths.

| Models | | UniTST (Ours) | | SOFT (2024) | | iTransformer (2023) | | RLinear (2023) | | PatchTST (2023) | | Crossformer (2023) | | TiDE (2023) | | TimesNet (2023) | | DLinear (2023) | | SCINet (2022a) | | FEDformer (2022) | | Stationary (2022b) | | Autoformer (2021) | |
|---|---|---|---|---|---|---|---|---|---|---|---|---|---|---|---|---|---|---|---|---|---|---|---|---|---|---|---|
| Metric | | MSE | MAE | MSE | MAE | MSE | MAE | MSE | MAE | MSE | MAE | MSE | MAE | MSE | MAE | MSE | MAE | MSE | MAE | MSE | MAE | MSE | MAE | MSE | MAE | MSE | MAE |
| ETTm1 | 96 | 0.313 | 0.352 | 0.325 | 0.361 | 0.334 | 0.368 | 0.355 | 0.376 | 0.329 | 0.367 | 0.404 | 0.426 | 0.364 | 0.387 | 0.338 | 0.375 | 0.345 | 0.372 | 0.418 | 0.438 | 0.379 | 0.419 | 0.386 | 0.398 | 0.505 | 0.475 |
| | 192 | 0.359 | 0.380 | 0.375 | 0.389 | 0.377 | 0.391 | 0.391 | 0.392 | 0.367 | 0.385 | 0.450 | 0.451 | 0.398 | 0.404 | 0.374 | 0.387 | 0.380 | 0.389 | 0.439 | 0.450 | 0.426 | 0.441 | 0.459 | 0.444 | 0.553 | 0.496 |
| | 336 | 0.395 | 0.404 | 0.405 | 0.412 | 0.426 | 0.420 | 0.424 | 0.415 | 0.399 | 0.410 | 0.532 | 0.515 | 0.428 | 0.425 | 0.410 | 0.411 | 0.413 | 0.413 | 0.490 | 0.485 | 0.445 | 0.459 | 0.495 | 0.464 | 0.621 | 0.537 |
| | 720 | 0.449 | 0.440 | 0.466 | 0.447 | 0.491 | 0.459 | 0.487 | 0.450 | 0.454 | 0.439 | 0.666 | 0.589 | 0.487 | 0.461 | 0.478 | 0.450 | 0.474 | 0.453 | 0.595 | 0.550 | 0.543 | 0.490 | 0.585 | 0.516 | 0.671 | 0.561 |
| | Avg | 0.379 | 0.394 | 0.393 | 0.403 | 0.407 | 0.410 | 0.414 | 0.407 | 0.387 | 0.400 | 0.513 | 0.496 | 0.419 | 0.419 | 0.400 | 0.406 | 0.403 | 0.407 | 0.485 | 0.481 | 0.448 | 0.452 | 0.481 | 0.456 | 0.588 | 0.517 |
| ETTm2 | 96 | 0.178 | 0.262 | 0.180 | 0.261 | 0.180 | 0.264 | 0.182 | 0.265 | 0.175 | 0.259 | 0.287 | 0.366 | 0.207 | 0.305 | 0.187 | 0.267 | 0.193 | 0.292 | 0.286 | 0.377 | 0.203 | 0.287 | 0.192 | 0.274 | 0.255 | 0.339 |
| | 192 | 0.243 | 0.304 | 0.246 | 0.306 | 0.250 | 0.309 | 0.246 | 0.304 | 0.241 | 0.302 | 0.414 | 0.492 | 0.290 | 0.364 | 0.249 | 0.309 | 0.284 | 0.362 | 0.399 | 0.445 | 0.269 | 0.328 | 0.280 | 0.339 | 0.281 | 0.340 |
| | 336 | 0.302 | 0.341 | 0.319 | 0.352 | 0.311 | 0.348 | 0.307 | 0.342 | 0.305 | 0.343 | 0.597 | 0.542 | 0.377 | 0.422 | 0.321 | 0.351 | 0.369 | 0.427 | 0.637 | 0.591 | 0.325 | 0.366 | 0.334 | 0.361 | 0.339 | 0.372 |
| | 720 | 0.398 | 0.395 | 0.405 | 0.401 | 0.412 | 0.407 | 0.407 | 0.398 | 0.402 | 0.400 | 1.730 | 1.042 | 0.558 | 0.524 | 0.408 | 0.403 | 0.554 | 0.522 | 0.960 | 0.735 | 0.421 | 0.415 | 0.417 | 0.413 | 0.433 | 0.432 |
| | Avg | 0.280 | 0.326 | 0.287 | 0.330 | 0.288 | 0.332 | 0.286 | 0.327 | 0.281 | 0.326 | 0.757 | 0.610 | 0.358 | 0.404 | 0.291 | 0.333 | 0.350 | 0.401 | 0.571 | 0.537 | 0.305 | 0.349 | 0.306 | 0.347 | 0.327 | 0.371 |
| ETTh1 | 96 | 0.383 | 0.398 | 0.381 | 0.399 | 0.386 | 0.405 | 0.386 | 0.395 | 0.414 | 0.419 | 0.423 | 0.448 | 0.479 | 0.464 | 0.384 | 0.402 | 0.386 | 0.400 | 0.654 | 0.599 | 0.376 | 0.419 | 0.513 | 0.491 | 0.449 | 0.459 |
| | 192 | 0.434 | 0.426 | 0.435 | 0.431 | 0.441 | 0.436 | 0.437 | 0.424 | 0.460 | 0.445 | 0.471 | 0.474 | 0.525 | 0.492 | 0.436 | 0.429 | 0.437 | 0.432 | 0.719 | 0.631 | 0.420 | 0.448 | 0.534 | 0.504 | 0.500 | 0.482 |
| | 336 | 0.471 | 0.445 | 0.480 | 0.452 | 0.487 | 0.458 | 0.479 | 0.446 | 0.501 | 0.466 | 0.570 | 0.546 | 0.565 | 0.515 | 0.491 | 0.469 | 0.481 | 0.459 | 0.778 | 0.659 | 0.459 | 0.465 | 0.588 | 0.535 | 0.521 | 0.496 |
| | 720 | 0.479 | 0.469 | 0.499 | 0.488 | 0.503 | 0.491 | 0.481 | 0.470 | 0.500 | 0.488 | 0.653 | 0.621 | 0.594 | 0.558 | 0.521 | 0.500 | 0.519 | 0.516 | 0.836 | 0.699 | 0.506 | 0.507 | 0.643 | 0.616 | 0.514 | 0.512 |
| | Avg | 0.442 | 0.435 | 0.449 | 0.442 | 0.454 | 0.447 | 0.446 | 0.434 | 0.469 | 0.454 | 0.529 | 0.522 | 0.541 | 0.507 | 0.458 | 0.450 | 0.456 | 0.452 | 0.747 | 0.647 | 0.440 | 0.460 | 0.570 | 0.537 | 0.496 | 0.487 |
| ETTh2 | 96 | 0.292 | 0.342 | 0.297 | 0.347 | 0.297 | 0.349 | 0.288 | 0.338 | 0.302 | 0.348 | 0.745 | 0.584 | 0.400 | 0.440 | 0.340 | 0.374 | 0.333 | 0.387 | 0.707 | 0.621 | 0.358 | 0.397 | 0.476 | 0.458 | 0.346 | 0.388 |
| | 192 | 0.370 | 0.390 | 0.373 | 0.394 | 0.380 | 0.400 | 0.374 | 0.390 | 0.388 | 0.400 | 0.877 | 0.656 | 0.528 | 0.509 | 0.402 | 0.414 | 0.477 | 0.476 | 0.860 | 0.689 | 0.429 | 0.439 | 0.512 | 0.493 | 0.456 | 0.452 |
| | 336 | 0.382 | 0.408 | 0.410 | 0.426 | 0.428 | 0.432 | 0.415 | 0.426 | 0.426 | 0.433 | 1.043 | 0.731 | 0.643 | 0.571 | 0.452 | 0.452 | 0.594 | 0.541 | 1.000 | 0.744 | 0.496 | 0.487 | 0.552 | 0.551 | 0.482 | 0.486 |
| | 720 | 0.409 | 0.431 | 0.411 | 0.433 | 0.427 | 0.445 | 0.420 | 0.440 | 0.431 | 0.446 | 1.104 | 0.763 | 0.874 | 0.679 | 0.462 | 0.468 | 0.831 | 0.657 | 1.249 | 0.838 | 0.463 | 0.474 | 0.562 | 0.560 | 0.515 | 0.511 |
| | Avg | 0.363 | 0.393 | 0.373 | 0.400 | 0.383 | 0.407 | 0.374 | 0.398 | 0.387 | 0.407 | 0.942 | 0.684 | 0.611 | 0.550 | 0.414 | 0.427 | 0.559 | 0.515 | 0.954 | 0.723 | 0.437 | 0.449 | 0.526 | 0.516 | 0.450 | 0.459 |
| ECL | 96 | 0.139 | 0.235 | 0.143 | 0.233 | 0.148 | 0.240 | 0.201 | 0.281 | 0.181 | 0.270 | 0.219 | 0.314 | 0.237 | 0.329 | 0.168 | 0.272 | 0.197 | 0.282 | 0.247 | 0.345 | 0.193 | 0.308 | 0.169 | 0.273 | 0.201 | 0.317 |
| | 192 | 0.155 | 0.250 | 0.158 | 0.248 | 0.162 | 0.253 | 0.201 | 0.283 | 0.188 | 0.274 | 0.231 | 0.322 | 0.236 | 0.330 | 0.184 | 0.289 | 0.196 | 0.285 | 0.257 | 0.355 | 0.201 | 0.315 | 0.182 | 0.286 | 0.222 | 0.334 |
| | 336 | 0.170 | 0.268 | 0.178 | 0.269 | 0.178 | 0.269 | 0.215 | 0.298 | 0.204 | 0.293 | 0.246 | 0.337 | 0.249 | 0.344 | 0.198 | 0.300 | 0.209 | 0.301 | 0.269 | 0.369 | 0.214 | 0.329 | 0.200 | 0.304 | 0.231 | 0.338 |
| | 720 | 0.198 | 0.293 | 0.218 | 0.305 | 0.225 | 0.317 | 0.257 | 0.331 | 0.246 | 0.324 | 0.280 | 0.363 | 0.284 | 0.373 | 0.220 | 0.320 | 0.245 | 0.333 | 0.299 | 0.390 | 0.246 | 0.355 | 0.222 | 0.321 | 0.254 | 0.361 |
| | Avg | 0.166 | 0.262 | 0.174 | 0.264 | 0.178 | 0.270 | 0.219 | 0.298 | 0.205 | 0.290 | 0.244 | 0.334 | 0.251 | 0.344 | 0.192 | 0.295 | 0.212 | 0.300 | 0.268 | 0.365 | 0.214 | 0.327 | 0.193 | 0.296 | 0.227 | 0.338 |
| Exchange | 96 | 0.080 | 0.198 | – | – | 0.086 | 0.206 | 0.093 | 0.217 | 0.088 | 0.205 | 0.256 | 0.367 | 0.094 | 0.218 | 0.107 | 0.234 | 0.088 | 0.218 | 0.267 | 0.396 | 0.148 | 0.278 | 0.111 | 0.237 | 0.197 | 0.323 |
| | 192 | 0.173 | 0.296 | – | – | 0.177 | 0.299 | 0.184 | 0.307 | 0.176 | 0.299 | 0.470 | 0.509 | 0.184 | 0.307 | 0.226 | 0.344 | 0.176 | 0.315 | 0.351 | 0.459 | 0.271 | 0.315 | 0.219 | 0.335 | 0.300 | 0.369 |
| | 336 | 0.314 | 0.406 | – | – | 0.331 | 0.417 | 0.351 | 0.432 | 0.301 | 0.397 | 1.268 | 0.883 | 0.349 | 0.431 | 0.367 | 0.448 | 0.313 | 0.427 | 1.324 | 0.853 | 0.460 | 0.427 | 0.421 | 0.476 | 0.509 | 0.524 |
| | 720 | 0.838 | 0.693 | – | – | 0.847 | 0.691 | 0.886 | 0.714 | 0.901 | 0.714 | 1.767 | 1.068 | 0.852 | 0.698 | 0.964 | 0.746 | 0.839 | 0.695 | 1.058 | 0.797 | 1.195 | 0.695 | 1.092 | 0.769 | 1.447 | 0.941 |
| | Avg | 0.351 | 0.398 | – | – | 0.360 | 0.403 | 0.378 | 0.417 | 0.367 | 0.404 | 0.940 | 0.707 | 0.370 | 0.413 | 0.416 | 0.443 | 0.354 | 0.414 | 0.750 | 0.626 | 0.519 | 0.429 | 0.461 | 0.454 | 0.613 | 0.539 |
| Traffic | 96 | 0.402 | 0.255 | 0.376 | 0.251 | 0.395 | 0.268 | 0.649 | 0.389 | 0.462 | 0.295 | 0.522 | 0.290 | 0.805 | 0.493 | 0.593 | 0.321 | 0.650 | 0.396 | 0.788 | 0.499 | 0.587 | 0.366 | 0.612 | 0.338 | 0.613 | 0.388 |
| | 192 | 0.426 | 0.268 | 0.398 | 0.261 | 0.417 | 0.276 | 0.601 | 0.366 | 0.466 | 0.296 | 0.530 | 0.293 | 0.756 | 0.474 | 0.617 | 0.336 | 0.598 | 0.370 | 0.789 | 0.505 | 0.604 | 0.373 | 0.613 | 0.340 | 0.616 | 0.382 |
| | 336 | 0.449 | 0.275 | 0.415 | 0.269 | 0.433 | 0.283 | 0.609 | 0.369 | 0.482 | 0.304 | 0.558 | 0.305 | 0.762 | 0.477 | 0.629 | 0.336 | 0.605 | 0.373 | 0.797 | 0.508 | 0.621 | 0.383 | 0.618 | 0.328 | 0.622 | 0.337 |
| | 720 | 0.489 | 0.297 | 0.447 | 0.287 | 0.467 | 0.302 | 0.647 | 0.387 | 0.514 | 0.322 | 0.589 | 0.328 | 0.719 | 0.449 | 0.640 | 0.350 | 0.645 | 0.394 | 0.841 | 0.523 | 0.626 | 0.382 | 0.653 | 0.355 | 0.660 | 0.408 |
| | Avg | 0.441 | 0.274 | 0.409 | 0.267 | 0.428 | 0.282 | 0.626 | 0.378 | 0.481 | 0.304 | 0.550 | 0.304 | 0.760 | 0.473 | 0.620 | 0.336 | 0.625 | 0.383 | 0.804 | 0.509 | 0.610 | 0.376 | 0.624 | 0.340 | 0.628 | 0.379 |
| Weather | 96 | 0.156 | 0.202 | 0.166 | 0.208 | 0.174 | 0.214 | 0.192 | 0.232 | 0.177 | 0.218 | 0.158 | 0.230 | 0.202 | 0.261 | 0.172 | 0.220 | 0.196 | 0.255 | 0.221 | 0.306 | 0.217 | 0.296 | 0.173 | 0.223 | 0.266 | 0.336 |
| | 192 | 0.207 | 0.250 | 0.217 | 0.253 | 0.221 | 0.254 | 0.240 | 0.271 | 0.225 | 0.259 | 0.206 | 0.277 | 0.242 | 0.298 | 0.219 | 0.261 | 0.237 | 0.296 | 0.261 | 0.340 | 0.276 | 0.336 | 0.245 | 0.285 | 0.307 | 0.367 |
| | 336 | 0.263 | 0.292 | 0.282 | 0.300 | 0.278 | 0.296 | 0.292 | 0.307 | 0.278 | 0.297 | 0.272 | 0.335 | 0.287 | 0.335 | 0.280 | 0.306 | 0.283 | 0.335 | 0.309 | 0.378 | 0.339 | 0.380 | 0.321 | 0.338 | 0.359 | 0.395 |
| | 720 | 0.340 | 0.341 | 0.356 | 0.351 | 0.358 | 0.347 | 0.364 | 0.353 | 0.354 | 0.348 | 0.398 | 0.418 | 0.351 | 0.386 | 0.365 | 0.359 | 0.345 | 0.381 | 0.377 | 0.427 | 0.403 | 0.428 | 0.414 | 0.410 | 0.419 | 0.428 |
| | Avg | 0.241 | 0.271 | 0.255 | 0.278 | 0.258 | 0.278 | 0.272 | 0.291 | 0.259 | 0.281 | 0.259 | 0.315 | 0.271 | 0.320 | 0.259 | 0.287 | 0.265 | 0.317 | 0.292 | 0.363 | 0.309 | 0.360 | 0.288 | 0.314 | 0.338 | 0.382 |
| Solar-Energy | 96 | 0.189 | 0.228 | 0.200 | 0.230 | 0.203 | 0.237 | 0.322 | 0.339 | 0.234 | 0.286 | 0.310 | 0.331 | 0.312 | 0.399 | 0.250 | 0.292 | 0.290 | 0.378 | 0.237 | 0.344 | 0.242 | 0.342 | 0.215 | 0.249 | 0.884 | 0.711 |
| | 192 | 0.222 | 0.253 | 0.229 | 0.253 | 0.233 | 0.261 | 0.359 | 0.356 | 0.267 | 0.310 | 0.734 | 0.725 | 0.339 | 0.416 | 0.296 | 0.318 | 0.320 | 0.398 | 0.280 | 0.380 | 0.285 | 0.380 | 0.254 | 0.272 | 0.834 | 0.692 |
| | 336 | 0.242 | 0.275 | 0.243 | 0.269 | 0.248 | 0.273 | 0.397 | 0.369 | 0.290 | 0.315 | 0.750 | 0.735 | 0.368 | 0.430 | 0.319 | 0.330 | 0.353 | 0.415 | 0.304 | 0.389 | 0.282 | 0.376 | 0.290 | 0.296 | 0.941 | 0.723 |
| | 720 | 0.247 | 0.282 | 0.245 | 0.272 | 0.249 | 0.275 | 0.397 | 0.356 | 0.289 | 0.317 | 0.769 | 0.765 | 0.370 | 0.425 | 0.338 | 0.337 | 0.356 | 0.413 | 0.308 | 0.388 | 0.357 | 0.427 | 0.285 | 0.295 | 0.882 | 0.717 |
| | Avg | 0.225 | 0.260 | 0.229 | 0.256 | 0.233 | 0.262 | 0.369 | 0.356 | 0.270 | 0.307 | 0.641 | 0.639 | 0.347 | 0.417 | 0.301 | 0.319 | 0.330 | 0.401 | 0.282 | 0.375 | 0.291 | 0.381 | 0.261 | 0.381 | 0.885 | 0.711 |
| 1st Count | | 30 | 27 | 6 | 11 | 2 | 1 | 1 | 5 | 3 | 5 | 1 | 0 | 0 | 0 | 0 | 0 | 0 | 0 | 0 | 0 | 4 | 0 | 0 | 0 | | |

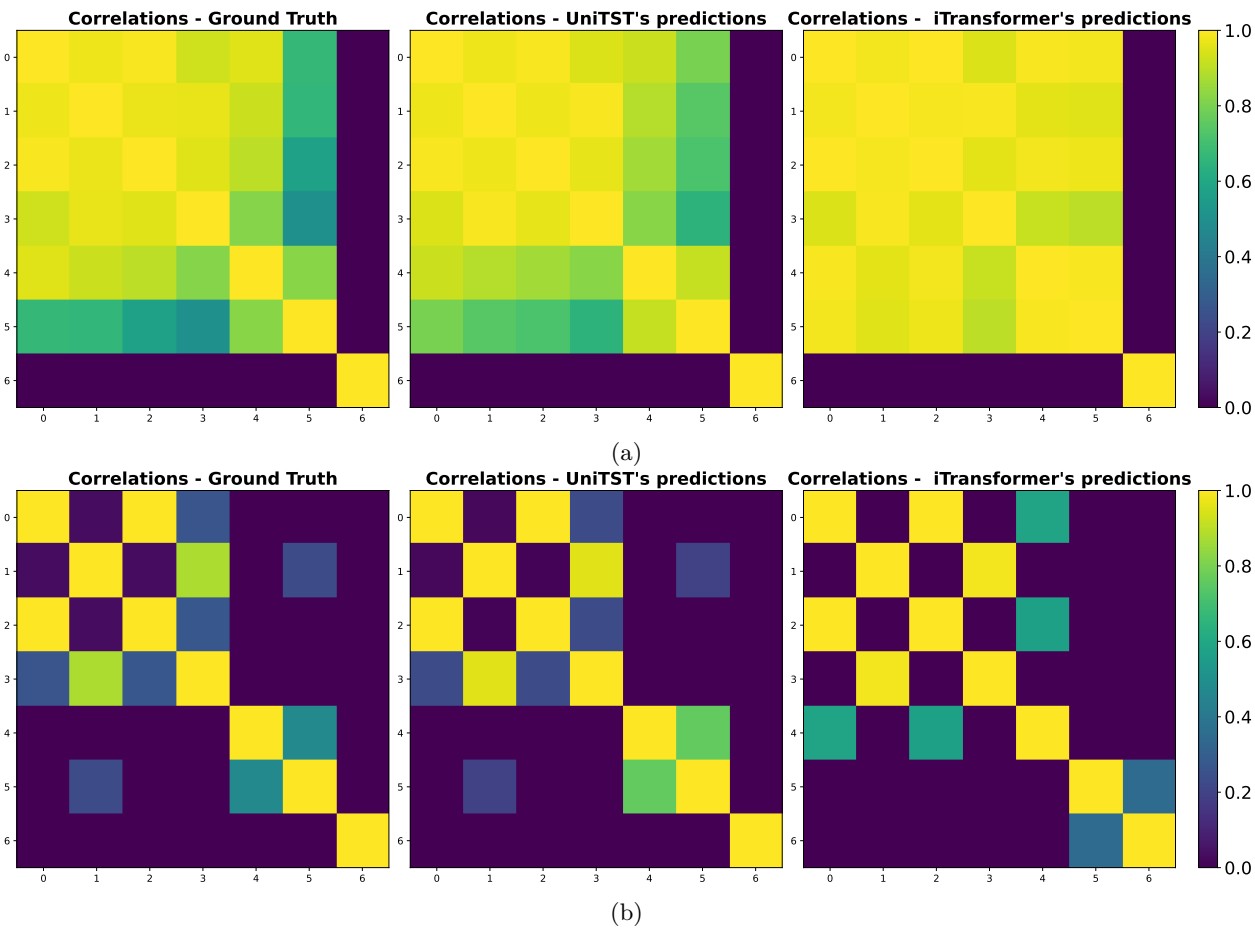

Figure 11: The correlation maps of multivariate relationship with different models on **ETTm1**. The x-axis and y-axis both represent the variate index. The darker color indicates the stronger correlation.

