# OpenReview forum: "UniTST: Effectively Modeling Inter-Series and Intra-Series Dependencies for Multivariate Time Series Forecasting"
_TMLR — Accepted by TMLR_

### Review · Reviewer_6jKS · 2025-03-25

**Summary Of Contributions:**

This paper introduces UniTST, a transformer-based architecture for multivariate time series forecasting that effectively addresses both inter-series and intra-series dependencies using a unified attention mechanism. The approach employs a dispatcher module with fixed query tokens to maintain efficiency when handling numerous variates. The model processes flattened patch tokens to simultaneously capture temporal and variate relationships, overcoming limitations in previous architectures.

**Audience:**

Yes

**Broader Impact Concerns:**

The paper presents a clean, efficient attention mechanism for multivariate time series forecasting that provides meaningful improvements over existing approaches, particularly with shorter lookback windows. The unified approach to capturing dependencies and the dispatcher mechanism represent valuable contributions to the time series and data mining field.

**Claims And Evidence:**

Yes

**Requested Changes:**

Please see the weaknesses.

**Strengths And Weaknesses:**

Strengths:
1. The unified attention mechanism offers a simple yet effective adaptation to existing multivariate Transformer models, enabling comprehensive modeling of complex dependencies.
2. The paper includes visual explanations (Figures 1 and 2) that effectively communicate the model architecture and underlying concepts.
3. The dispatcher mechanism significantly reduces computational complexity compared to full pairwise interactions across all variates.
4. Comprehensive experiments across 13 diverse datasets demonstrate consistent performance improvements over state-of-the-art models like iTransformer, particularly for lookback length of 96.

Limitations:
1. The performance advantage over iTransformer diminishes with larger lookback lengths (as shown in Figure 5), suggesting that UniTST may not improve upon the state-of-the-art when using optimal lookback lengths.
2. The argument that time-wise and variate-wise attention must be integrated within the same mechanism could be more thoroughly substantiated, as alternating layers of time-wise and variate-wise attention might theoretically achieve similar information flow.

Suggestions for Improvement:
1. Include computational efficiency comparisons (training times) between UniTST, Crossformer, and iTransformer to better highlight the practical benefits of the dispatcher mechanism.
2. Explore the interpretability aspects of the dispatcher module to potentially extract meaningful patterns about time series relationships.
3. Consider investigating a more adaptive architecture where attention is primarily used in the time direction, with variable interactions occurring only when necessary through a dynamic dispatcher, which could further improve scalability for massive but sparsely interacting systems.

---

> ### Author Response · Authors · 2025-04-22
>
> Dear Review 6jKS,
>
> Thanks for the valuable suggestions. I'm terribly sorry for the delay in responding. I got a hand injury a few weeks ago that makes typing very difficult. We post our response as follows:
>
> > The performance advantage over iTransformer diminishes with larger lookback lengths
>
> We agree that when lookback lengths increase the performance advantage is not that significant. However, we'd like to point out that in real world cases, most of time we don't have the sufficient lookback historical data. In those case, the ability to perform well becomes very important. In Figure 5, we also show that we can achieve best performance compared with others when we only use 48 as lookback.
>
>
> > We provide the computational complexity analysis of different models here.
>
> As Feedforward networks in different models have similar complexities, we mainly consider analysing the computational complexity of the attention mechanism as shown below:
>
> 1. **UniTST**: $\mathcal{O}(kNp)$ where $k$ is the number of dispatchers, $N$ is the number of variates, and $p$ is the number of patches within a variate. Cross-attention with dispatchers to reduce the complexity.
> 2. **iTransformer**: $\mathcal{O}(N^2)$ where $N$ is the number of variates. Self-attention on the variate dimension.
> 3. **PatchTST**: $\mathcal{O}(Np^2)$ where $N$ is the number of variates and $p$ is the number of patches. Self-attention on the time dimension.
>
> We can see that different models have different advantages in different scenarios. For example, when handling data with long time series but fewer variates, iTransformer should be faster than others as it doesn't depend on $p$. Comparing UniTST and PatchTST, when $p$ is relatively small, then the complexity should be similar (we set $k$ as 10 in our experiments). UniTST may be slower than iTransformer when the length of the time series and the number of variates are both extremely large.
>
>
> We also provide the training time as follows:
>
> Training Time: (second/iteration)
>
> | Dataset   | UniTST (ours) | iTransformer | PatchTST |
> |-----------|----------------|--------------|----------|
> | Exchange  | 0.0678         | 0.0196       | 0.0504   |
> | Weather   | 0.0492         | 0.1034       | 0.0647   |
>
>  Compared with iTransformer and PatchTST, UniTST is sometimes faster for training. The complexity increment mainly comes from the increment of the number of "tokens" as our method applies attention on flattened sequences.
>
> > alternating layers of time-wise and variate-wise attention might theoretically achieve similar information flow.
>
> We agree that it might be the case theoretically to some extent. However, that might be difficult to achieve empirically, and we believe that our proposed method can be a more direct method to model inter-series and intra-series dependencies.
>
> > Explore the interpretability aspects of the dispatcher module to potentially extract meaningful patterns about time series relationships.
>
> We provide the case study in Figure 9 to show the Multivariate Correlations captured by our model
>
> > Consider investigating a more adaptive architecture where attention is primarily used in the time direction, with variable interactions occurring only when necessary through a dynamic dispatcher, which could further improve scalability for massive but sparsely interacting systems.
>
> Thanks for the suggestion. We will consider this as future work.

---

### Review · Reviewer_yYJE · 2025-04-03

**Summary Of Contributions:**

This paper proposes a novel Transformer model named UniTST for multivariate time series forecasting (MTSF). The key contributions include:

Problem Analysis: The authors identify the limitations of existing Transformer models in simultaneously capturing inter-variate and intra-variate dependencies, and validate the importance of these dependencies through real-world data.

Model Design: To address the increased computational cost, the authors propose a unified attention mechanism that flattens patches from different variates into a single sequence to directly model cross-variate and cross-time dependencies. Additionally, a dispatcher module is introduced to reduce computational complexity from quadratic to linear, significantly improving scalability.

**Audience:**

Yes

**Broader Impact Concerns:**

The paper does not raise significant ethical concerns.

**Claims And Evidence:**

Yes

**Requested Changes:**

Additional Comparative Experiments: Include comparisons with state-of-the-art methods like SOFTS to comprehensively demonstrate the model's advantages.

Enhanced Interpretability: Provide case studies or theoretical analysis to clarify how the dispatcher selects and prioritizes dependencies.

**Strengths And Weaknesses:**

Strengths:

Innovation: The unified attention mechanism and dispatcher module present a novel solution to the challenge of modeling both inter-variate and intra-variate dependencies simultaneously.

Theoretical Foundation: The introduction of the cross-time cross-variate correlation coefficient provides a theoretical basis for the diversity of dependencies, supporting the model's design.

Weaknesses:

Interpretability: While the attention weights are visualized to illustrate dependency relationships, the paper lacks a detailed explanation of how the dispatcher allocates information. A deeper analysis of the mechanism's workings would strengthen the study.

Baseline Comparison: The paper does not compare with recent related work, such as SOFTS: Efficient Multivariate Time Series Forecasting with Series-Core Fusion (NeurIPS 2024). Additional experiments are needed to further validate the model's superiority.

---

> ### Author Response · Authors · 2025-05-02
>
> Dear reviewer yYJE,
>
> Thanks for the valuable suggestions! I'm terribly sorry for the delay in responding. We post our response as follows:
>
> > Additional Comparative Experiments: Include comparisons with state-of-the-art methods like SOFTS to comprehensively demonstrate the model's advantages.
>
> We have revised the manuscript and included the comparison with SOFTS in Table 1, 2, and 6.
>
> > Enhanced Interpretability: Provide case studies or theoretical analysis to clarify how the dispatcher selects and prioritizes dependencies.
>
> Thanks for the great advice. For the interpretability,  in the original draft (Figure 7), we show how dispatchers selects and prioritizes dependencies by showing the attention weights between two patch tokens and how the distributions of it changes on different layers. It indicates that dispatchers in the deepers layers can learn how to distribute the information to important tokens.
>
> For specific case studies and theoretical analysis, due to the time limit and my physical disability at the moment, we might be only able to investigate later.

---

### Review · Reviewer_RSTB · 2025-04-06

**Summary Of Contributions:**

This paper proposes UniTST for time series forecasting, which can simultaneously capture both inter-variable and intra-variable dependencies. Experiments on multiple real-world datasets demonstrate that UniTST achieves superior performance.

**Audience:**

Yes

**Claims And Evidence:**

Yes

**Requested Changes:**

As in Cons.

**Strengths And Weaknesses:**

Pros:

1. The paper discusses the limitations of previous Transformer-based methods for multivariate time series forecasting (inability to capture both inter-variable and intra-variable relationships simultaneously) and validates this viewpoint through examples.
2. UniTST effectively captures both inter-variable and intra-variable dependencies by flattening all patches from different variables into a unified sequence, and reduces memory overhead through the dispatches module. Ablation study demonstrate the effectiveness of each module.
3. Extensive experiments on multiple real-world datasets have demonstrated the effectiveness of the method.

Cons:

1. In the rapidly evolving field of time series forecasting, it is insufficient to only use iTransformer as a baseline. Further comparisons with more advanced methods (in 2024 or later) are needed to demonstrate the effectiveness of the approach, such as Samformer [1], ElasTST [2], and DeformableTST [3].
2. Although the effectiveness of the dispatcher module has been explored, demonstrating its ability to effectively reduce memory consumption, an analysis of the model's efficiency remains essential.
3. As seen in Table 4, there is no clear relationship between the number of dispatcher modules and the MSE performance across different datasets. Does this imply that the number of dispatcher modules is a sensitive hyperparameter, requiring fine-tuning across different datasets to identify the most suitable value?
4. UniTST captures both inter-variable and intra-variable dependencies by flattening all patches into a unified sequence. However, different datasets have varying degrees of dependence between variables and within variables. For example, datasets like ECL, which have a large number of channels, may require a greater emphasis on capturing inter-variable relationships. It is necessary to further complement the ablation experiments to evaluate the performance of UniTST compared to models that only model intra-variable relationships or inter-variable relationships on different datasets.
5. For the inter-variable relationship visualization in Figure 9, it is necessary to provide results on additional datasets to demonstrate the effective capture of inter-variable relationships by UniTST.

[1] Samformer: Unlocking the potential of transformers in time series forecasting with sharpness-aware minimization and channel-wise attention. ICML, 2024.

[2] ElasTST: Towards Robust Varied-Horizon Forecasting with Elastic Time-Series Transformer. NeurIPS, 2024.

[3] DeformableTST: Transformer for Time Series Forecasting without Over-reliance on Patching. NeurIPS, 2024.

---

> ### Author Response · Authors · 2025-04-30
>
> Dear Reviewer RSTB:
>
> Thanks for the valuable suggestions. I am terribly sorry for the delay in responding. I got a hand injury a few weeks ago that makes typing very difficult. We tried our best to update the revised manuscript and we post our respose as follows:
>
> > Further comparisons with more advanced methods (in 2024 or later)
>
> We have added SOFT [1] as an additional baseline in our revised manuscript, which can be viewed on Table 2 and 3. Due the time limit and the personal physical disability at the moment, we do not include more baselines. However, we mentioned the ones you pointed out in the related work to help people better understand the current literature in the field.
>
> > Although the effectiveness of the dispatcher module has been explored, demonstrating its ability to effectively reduce memory consumption, an analysis of the model's efficiency remains essential.
>
> We agree that the model efficiency analysis is important. We provided the discussion on computational complexity on Appendix B.
> We also provide the emipirical training time as follows:
>
> Training Time: (second/iteration)
>
> | Dataset   | UniTST (ours) | iTransformer | PatchTST |
> |-----------|----------------|--------------|----------|
> | Exchange  | 0.0678         | 0.0196       | 0.0504   |
> | Weather   | 0.0492         | 0.1034       | 0.0647   |
>
> Compared with iTransformer and PatchTST, UniTST is sometimes faster for training. The complexity increment mainly comes from the increment of the number of "tokens" as our method applies attention on flattened sequences.
>
>
> > As seen in Table 4, there is no clear relationship between the number of dispatcher modules and the MSE performance across different datasets. Does this imply that the number of dispatcher modules is a sensitive hyperparameter, requiring fine-tuning across different datasets to identify the most suitable value?
>
> We agree that the number of dispatcheres is a hyperparameter. However, we believe it is not very sensitive and it doesn't require massive tuning. As shown in Table 4, varying the number of dispatchers provides MSE from 0.155 to 0.157 for Weather dataset and from 0.133 to 0.134 for ECL, which may not be a significant difference. We believe that 10 or 20 is a good start to try with, if users want to tune this hyperparmeter.
>
> > UniTST captures both inter-variable and intra-variable dependencies by flattening all patches into a unified sequence. However, different datasets have varying degrees of dependence between variables and within variables. For example, datasets like ECL, which have a large number of channels, may require a greater emphasis on capturing inter-variable relationships. It is necessary to further complement the ablation experiments to evaluate the performance of UniTST compared to models that only model intra-variable relationships or inter-variable relationships on different datasets.
>
> Besides ECL dataset, in Figure 5, we also use weather (containing only 21 variates) to compare UniTST with methods that only model intra-variable relationships (i.e., iTransformer) and that only model inter-variable relationships (i.e., PatchTST) on different lookback length
>
> > For the inter-variable relationship visualization in Figure 9, it is necessary to provide results on additional datasets to demonstrate the effective capture of inter-variable relationships by UniTST.
>
> Thanks for the valuable suggestion. We have provided the inter-variable relationship visualization on ETTm1 dataset in Figure 11 (Appendix C.3.2) as the additional evidence. It shows that the viriate dependencies are well-captured by our model.
>
>
> Reference:
>
> [1]  SOFTS: Efficient Multivariate Time Series Forecasting with Series-Core Fusion (NeurIPS 2024)

---

### Decision · Action_Editor_fL5E · 2025-06-01

**Recommendation:** Accept as is

**Comment:**

All reviewers positively acknowledge the significance and clarity of the paper. Reviewer yYJE highlights the paper’s comprehensive revisions, notably the inclusion of comparisons with state-of-the-art methods such as SOFTS and improvements in the interpretability of the dispatcher module through detailed attention visualizations. Reviewer 6jKS similarly appreciates the clarity, innovation, and empirical robustness of the proposed method. Reviewer RSTB, while leaning slightly towards acceptance, expresses mild reservations regarding the authors’ handling of baseline comparisons, noting that several suggested baselines were not fully incorporated. Despite these concerns, the reviewer concedes that the authors have adequately addressed major issues, bringing the paper to a technically sound and publishable state.

The revisions have sufficiently addressed the reviewers' core concerns, particularly regarding benchmark comparisons and interpretability.

**Audience:**

This paper is highly relevant to TMLR’s audience, particularly researchers and practitioners focused on time series forecasting, Transformer models, and deep learning methodologies. The novel unified attention mechanism and dispatcher module introduced by the authors are likely to attract considerable interest due to their practical utility and scalability in complex multivariate scenarios.

**Claims And Evidence:**

The reviewers agree that the claims presented in the paper are supported by accurate, clear, and convincing evidence. The proposed Transformer-based model, UniTST, addresses critical limitations in existing multivariate time series forecasting (MTSF) methods by effectively modeling inter-series and intra-series dependencies. The authors provide substantial justification through the cross-time cross-variate correlation coefficient, complemented by extensive experimental validations that demonstrate the model's efficacy on various real-world datasets.